



# Total OH reactivity over the Amazon rainforest: variability with temperature, wind, rain, altitude, time of day, season, and an overall budget closure

Eva Y. Pfannerstill[1], Nina G. Reijrink[1,2], Achim Edtbauer[1], Akima Ringsdorf[1], Nora Zannoni[1], Alessandro Araújo[3], Florian Ditas[1,a], Bruna A. Holanda[1], Marta O. Sá[4], Anywhere Tsokankunku[1], David Walter[1], Stefan Wolff[1], Jošt V. Lavrič[5], Christopher Pöhlker[1], Matthias Sörgel[1], Jonathan Williams[1,6]

[1]Atmospheric Chemistry and Multiphase Chemistry Departments, Max Planck Institute for Chemistry, 55128 Mainz, Germany
[2]Département Sciences de l'Atmosphère et Génie de l'Environnement (SAGE), IMT Lille Douai, 59508 Douai, France
[3]Empresa Brasileira de Pesquisa Agropecuária (Embrapa) Amazonia Oriental, CEP 66095-100, Belém, Brazil
[4]Instituto Nacional de Pesquisas da Amazônia (INPA), CEP 69067-375, Manaus, Brazil
[5]Biogeochemical Processes Department, Max Planck Institute for Biogeochemistry, 07745 Jena, Germany
[6]Energy, Environment and Water Research Center, The Cyprus Institute, 1645 Nicosia, Cyprus
[a]now at: Hessisches Landesamt für Naturschutz, Umwelt und Geologie, 65203 Wiesbaden, Germany

*Correspondence to*: Eva Y. Pfannerstill (eva.pfannerstill@mpic.de)

**Abstract.** The tropical forests are Earth's largest source of biogenic volatile organic compounds (BVOCs) and thus also the largest atmospheric sink region for the hydroxyl radical (OH). However, the OH sink above tropical forests is poorly understood, as past studies revealed large unattributed fractions of total OH reactivity. We present the first total OH reactivity and VOC measurements made at the Amazon Tall Tower Observatory (ATTO) at 80, 150, and 320 m above ground level, covering two dry seasons, one wet and one transition season in 2018–2019. By considering a wide range of previously unaccounted for VOCs, which we identified by PTR-ToF-MS, the unattributed fraction was with an overall average of 19 % within the measurement uncertainty of ~35 %. In terms of seasonal average OH reactivity, isoprene accounted for 23–43 % the total, oxygenated VOCs (OVOCs) for 22–40 %, while monoterpenes, sesquiterpenes, and green leaf volatiles combined were responsible for 9–14 %. These findings show that OVOCs were until now an underestimated contributor to the OH sink above the Amazon forest.

By day, total OH reactivity decreased towards higher altitudes with strongest vertical gradients observed around noon during the dry season (-0.026 $s^{-1}$ $m^{-1}$), while the gradient was inverted at night. Seasonal differences in total OH reactivity were observed, with the lowest daytime average and standard deviation of $19.9 \pm 6.2$ $s^{-1}$ during a wet–dry transition season with frequent precipitation, $23.7 \pm 6.5$ $s^{-1}$ during the wet season, and the highest average OH reactivities during two dry season observation periods with $28.1 \pm 7.9$ $s^{-1}$ and $29.1 \pm 10.8$ $s^{-1}$, respectively. The effects of different environmental parameters on the OH sink were investigated, and quantified, where possible. Precipitation caused short-term spikes in total OH reactivity, which were followed by below-normal OH reactivity for several hours. Biomass burning increased total OH reactivity by 2.7 $s^{-1}$ to 9.5 $s^{-1}$. We present a temperature-dependent parameterization of OH reactivity that could be applied in future models of the OH sink to further reduce our knowledge gaps in tropical forest OH chemistry.





## 1 Introduction

The Amazon rainforest, with its area of over 5.8 million km², contains more than half of the Earth's tropical forests (Morley, 2000), a quarter of global biodiversity (Dirzo and Raven, 2003) and nearly 15% of terrestrial biomass (Houghton et al., 2001; Bar-On et al., 2018). The Amazon biome is thus of global relevance for cycling of water, energy, carbon and nutrients (Andreae, 2002; Malhi, 2002; Werth, 2002; Aragão et al., 2014). Understanding its interaction with the atmosphere is highly important for understanding global biogeochemical cycles.

The emission of biogenic volatile organic compounds (BVOCs) by vegetation plays a fundamental role in atmospheric chemistry. Global BVOC emissions are, according to recent estimates, a factor of $\approx 6$ higher than anthropogenic VOC emissions (Guenther et al., 2012; Sindelarova et al., 2014; Hoesly et al., 2018). Tropical forests are Earth's largest BVOC source, contributing 65 % of the global isoprene emissions and 77 % of the global monoterpene emissions (Guenther et al.,

2012; Sindelarova et al., 2014). The relevance of BVOCs for atmospheric processes stems from their high chemical reactivity. Once released to the atmosphere, BVOCs undergo oxidation reactions within seconds to days, mainly reacting with OH radicals, which are formed during daytime from the ozone photoproduct $O^1D$ and water as well as through recycling reactions (Taraborrelli et al., 2012). This oxidation pathway influences regional tropospheric ozone and secondary organic aerosol formation (Palm et al., 2018; Wyche et al., 2014; Hamilton et al., 2013; Goto et al., 2008; Schulz et al., 2018), thereby

impacting oxidative stress to ecosystems, as well as cloud formation and global climate (Bates and Jacob, 2019; Scott et al., 2018; Engelhart et al., 2011; Pöschl et al., 2010; Heald and Spracklen, 2015). Additionally, the reaction of BVOCs with OH affects the regional atmospheric oxidation capacity, which in turn controls the residence times of longer-lived greenhouse gases (e.g. $CH_4$) and pollutants (e.g. CO) (Bates and Jacob, 2019; Arneth et al., 2010; Peñuelas and Staudt, 2010).

More than 30 000 different BVOCs are thought to be released from vegetation (e.g. Harley, 2013) including a wide range of chemical classes, such as terpenoids, alkenes, alcohols, aldehydes, and ketones (Guenther et al., 2012). One measurement survey found 264 BVOC species emitted from tropical trees in French Guyana (Courtois et al., 2009). However, for technical reasons, most studies of BVOCs in tropical forests to date have been limited to a modest number of known, abundant species such as isoprene, monoterpenes and few oxygenated compounds (e.g. Yáñez-Serrano et al., 2015; Langford et al., 2010; Rizzo

et al., 2010; Kuhn et al., 2007; Karl et al., 2007; Saxton et al., 2007; Rottenberger et al., 2004; Rinne et al., 2002; Kesselmeier et al., 2000).

Therefore, a measure of the combined effect of all VOCs and other OH reactive species in ambient air is necessary for a complete understanding of local atmospheric chemistry and of the OH sink. This directly measurable quantity is called total

OH reactivity, and its value determines the OH loss frequency in s$^{-1}$. By comparing to the OH reactivity calculated from all



individually measured compounds (termed the speciated OH reactivity), total OH reactivity can reveal the presence of VOCs that were not measured (termed the unattributed or "missing" OH reactivity).

The tropical forests are the regions of the world with the largest total OH reactivities in the air, with modelled annual averages of 30–50 s$^{-1}$ (Ferracci et al., 2018; Safieddine et al., 2017). Total OH reactivity in Amazonia is thought to be dominated by isoprene and other reactive BVOCs (Nölscher et al., 2016; Safieddine et al., 2017), which are emitted as a function of light and temperature (e.g. Jardine et al., 2015; Kuhn et al., 2004a; Rinne et al., 2002). However, a significant part of tropical rainforest OH reactivity remains unexplained by speciated trace gases measurements (Nölscher et al., 2016; Williams et al., 2016; Edwards et al., 2013) and by usual global atmospheric chemistry models (Ferracci et al., 2018) with unattributed fractions of 5–15 % in the wet and 79 % in the dry season (Nölscher et al., 2016). The unattributed fraction of OH reactivity in tropical forests is assumed to be due to unmeasured oxygenated intermediates (Edwards et al., 2013) or to a combination of OVOCs and multiple primary BVOCs (Nölscher et al., 2016). Atmospheric chemistry models usually do not consider this additional, unattributed OH sink, with significant implications for the lifetime of the hydroxyl radical and of methane (Ferracci et al., 2018).

In this work, we used comprehensive VOC information, including compounds with uncertain chemical structure measured by PTR-ToF-MS, for comparison with total OH reactivity for the first time at a tropical rainforest site. The only previous Amazon rainforest reactivity budget exercise was reported by Nölscher et al. (2016) from a smaller 80 m tower. It was based on data from a quadrupole PTR-MS. The PTR-ToF-MS system used in this study is more sensitive and had significantly higher mass resolution allowing more compounds to be measured. In this way, we have attempted to obtain closure on apportioning OH reactivity and to identify the previously unidentified OH sinks. Presented here are the first measurements of total OH reactivity at the 320 m high tower at the remote Amazon rainforest site ATTO (Amazon Tall Tower Observatory), covering one wet season, one wet-dry transition and two dry seasons in 2018–2019. We discuss vertical gradients between 80 m and 320 m above the rainforest floor, as well as seasonality and diel cycles of total OH reactivity and its composition. The influence of environmental parameters such as precipitation and biomass burning is investigated, and a seasonal parameterization for total OH reactivity dependence on temperature is developed.

## 2 Materials and Methods

### 2.1 Study site – Amazon Tall Tower Observatory

The Amazon Tall Tower Observatory (Andreae et al., 2015) is situated ~ 150 km northeast of Manaus, Brazil, in a dense, non-flooded *terra firme* (plateau) forest at ~ 120 m above sea level. The average rainfall at the site reaches its monthly maximum of ~ 335 mm in the wet (February to May) and its minimum of ~ 47 mm in the dry season (August to November) (Pöhlker et al., 2019). The wet–dry transition season covers June to July and the dry–wet transition season December to January (Andreae et al., 2015; Pöhlker et al., 2016). The main wind directions at the site are northeast during the wet season and east during the





dry season (Pöhlker et al., 2019). The site is equipped with several towers. The tall tower with a total height above ground level (a. g. l.) of 325 m is located at S 02°08.602′, W 59°00.033′. It has been in operation since 2018 with continuous

measurements of VOCs, ozone, particles and basic meteorological data. During intensive observation periods, these were complemented by total OH reactivity observations and sampling of VOCs for offline analysis. A walk–up tower (height a. g. l.: 80 m) located ~ 1 km from the tall tower is equipped with instrumentation for continuous measurements of meteorology, micrometeorology, greenhouse gases, phenology, inorganic trace gases, and aerosol particles since 2012. Maximum canopy top height of the trees surrounding the towers is ca. 35 m.


### 2.2 Measurement periods

Continuous total OH reactivity measurements were conducted in March 2018 (7 days; 09/03/2018–15/03/2018), October 2018 (6 days; 20/10/2018–26/10/2018), June 2019 (13 days; 10/06/2019–23/06/2019) and September 2019 (5 days; 22/09/2019–26/09/2019). These campaigns cover several seasons with March in the wet season (median soil moisture at 10 cm depth

0.169 $m^3$ $m^{-3}$), September and October in the dry season (median soil moistures 0.160 $m^3$ $m^{-3}$ and 0.161 $m^3$ $m^{-3}$, respectively). June is part of the wet–dry transition season. However, June 2019 was particularly wet with flooding at the riverbanks in the study region (median soil moisture 0.218 $m^3$ $m^{-3}$, see Fig. S1).

### 2.3 Sampling

Total OH reactivity, ozone and VOC measurement devices shared an inlet system. Measurement heights at the ATTO tall tower were 80, 150, and 320 m a. g. l., each equipped with its own, constantly flushed (15 L $min^{-1}$) inlet tube which brought air from aloft to a temperature-controlled laboratory container located adjacent to the tower. At the entry of each inlet, a regularly changed polytetrafluoroethylene (PTFE) filter (5 µm pore size) prevented contamination by particles or insects. The non-transparent inlet tubes (outer diameter: 0.5 inch = 1.27 cm, material: fluorinated ethylene propylene) were heated

(≈ 45 °C) and insulated with flexible elastomeric foam covering. A valve system was constructed to switch between the different heights, beginning with the lowest height at the full hour and switching to the next height every 5 min. Thus, each height was sampled four times an hour. In order to avoid underpressure at the end of the long inlet lines, the laboratory container was downstream of a chemically inert pump with a Teflon membrane (Neuberger KNF, Freiburg, Germany) with a sampling flow of ca. 17 L $min^{-1}$. The inlet residence time from 320 m was ≈ 80 s. The inlet response time, i.e. the time for all species to

reach a steady signal, determined by sampling a calibration gas mixture introduced at the 320 m inlet entrance, ranged between 88 s (for methanol) and 96 s (for acetaldehyde, benzene, trimethylbenzene, methacrolein), with an average of 93 s. Consequently, the data measured in the first 100 s after each valve switch were not included in the data analysis. Inlet losses from the 320 m inlet to the instrument ranged between 11 % and 30 % for all substances in the calibration mixture. We did not correct final data for inlet losses in order to keep VOC and total OH reactivity data, which were both measured from the same



inlet, comparable (inlet loss correction for total OH reactivity would change according to its composition, which is not entirely known), and because loss fractions for substances not included in the calibration mixture are not known.

## 2.4 Total OH reactivity, trace gas, and black carbon measurements

During all four campaigns, total OH reactivity was determined using the Comparative Reactivity Method (CRM, Sinha et al., 2008). The method was recently compared to other OH reactivity measurement devices (Fuchs et al., 2017) and has been

applied in various regions of the world (Yang et al., 2016). Briefly, CRM uses the known OH reactivity of a pyrrole gas standard (Westfalen AG, Münster, Germany) and compares it with the reactivity of all compounds found in ambient air in a competitive reaction. OH radicals are created inside a glass reactor by flushing humidified nitrogen (6.0 grade, Westfalen AG, Münster, Germany) over a Hg/Ar UV lamp (LOT Quantum Design, Darmstadt, Germany). CRM uses three different modes: C1 (OH scavenger + pyrrole + UV light at ≈ ambient humidity), C2 (OH + pyrrole, ambient humidity), and C3 level (ambient

air + pyrrole + OH). For a more detailed description, see Sinha et al. (2008) and Michoud et al. (2015).

In our study, the C2 level was measured for 15 min at the beginning of each hour, followed by 45 min of C3 measurement. C1 level determinations and calibrations with a pyrrole gas standard (Westfalen AG, Münster, Germany) were performed at least twice during each campaign. The C1 level was typically $68 \pm 1$ ppb (parts per billion = nmol mol$^{-1}$). The system was operated at a pyrrole/OH ratio of $2.76 \pm 0.23$ (average ± standard deviation). As is typical in the CRM method, this ratio deviates from

the pseudo-first order conditions assumed in the CRM equation (Sinha et al., 2008). Therefore, a correction had to be applied (see Sect. 2.5).

The pyrrole mixing ratio was monitored by a Proton Transfer Reaction–Quadrupole Mass Spectrometer (PTR–QMS, Ionicon Analytik, Innsbruck, Austria; Lindinger et al., 1998) at m/z = 68. The instrument was operated at 60 °C drift temperature, 2.2 mbar drift pressure, 600 V drift voltage, and 137 Td.

Simultaneously, VOC measurements up to m/z ≈ 300 were conducted using a PTR–Time of flight (ToF)–MS (Ionicon Analytik, Innsbruck, Austria; Jordan et al., 2009), operated at 60 °C drift temperature, 3.5 mbar drift pressure, 850 V drift voltage, and 120 Td. The instrument was calibrated with a gravimetrically prepared multicomponent VOC standard (Apel-Riemer Environmental Inc., Colorado, USA) once during each campaign. Diiodobenzene was continuously fed into the sample stream for mass scale calibration. The time resolution of the measurement was 20 s. Mass resolution (full width at half

maximum) ranged between 3000 and ~3500. A complete list of the trace gases measured by PTR–ToF–MS can be found in Table S1.

CO and methane were sampled at the walk-up tower at 80 m a. g. l using a cavity ring-down spectroscopy instrument (Picarro Inc., Santa Clara, USA). From preliminary CO and methane measurements at the 320 m tower, vertical gradients were taken to calculate approximate concentrations for 150 m and 320 m from the 80 m observations. For CO, 10 ppb and 30 ppb were

subtracted from the 80 m values to derive lower limit mixing ratios for 150 m and 320 m, respectively. Methane values measured at 80 m were assumed to be valid for all heights, because the small height-related differences observed are not in an order of magnitude that affects OH reactivity (Botía et al., 2020).



For monoterpene and sesquiterpene speciation, air samples were collected on adsorbent filled tubes equipped with ozone scrubbers at different heights on the ATTO tower (Zannoni et al., 2020a). Sampling occurred every three hours for two weeks at 80 m and 150 m (March 2018) and every hour for three days at 80 m, 150 m and 320 m (October 2018). Samples were analyzed in the laboratory by a TD-GC-ToF-MS (Thermodesorption-Gas Chromatographer-Time of Flight-Mass Spectrometer; Markes International, Llantrisant, United Kingdom).

Measurements of black carbon mass concentrations were obtained at the 325 m inlet of the ATTO tower using a multi-angle absorption photometer (MAAP, model 5012, Thermo Fisher Scientific, Waltham, USA), as described in Holanda et al. (2020).

## 2.5 Total OH reactivity data analysis

CRM data analysis and corrections were conducted following the procedures described in Pfannerstill et al. (2019). The humidity correction amounted to an average of 5 %, the ozone correction to an average of 2 % of the measured OH reactivity. An $NO/NO_2$ correction was not performed due to the low $NO_x$ levels at the site (Pfannerstill et al., 2018; Wolff, 2015) with maximum NO and $NO_2$ mixing ratios of 0.36 ppb and 0.62 ppb in the dry season (October 2018), respectively. The correction for pseudo–1st–order deviation was derived from tests using propene, isoprene and propane, resulting in a weighted correction factor in dependence of alkene fraction in calculated OH reactivity and pyrrole/OH ratio (Eq. 1):

$$F = F_1 X_{alkenes} + F_2 (1 - X_{alkenes}),\tag{1}$$

where $F$ is the total correction factor and the weighted, pyrrole/OH dependent correction factors were $F_1$ = -0.53*pyrrole/OH + 2.98 and $F_2$ = 0.63 for March 2018, $F_1$ = -1.20*pyrrole/OH + 4.73 and $F_2$ = -0.20*pyrrole/OH + 1.55 for October 2018, $F_1$ = -1.1*pyrrole/OH +4.50 and $F_2$ = 0.94 for June 2019, -0.52*pyrrole/OH +3.05 and $F_2$ = 0.94 for September 2019. $X_{alkenes}$ is the speciated OH reactivity fraction of alkenes. This correction increased OH reactivity by a factor of 1.12 ± 0.10 (average ± standard deviation).

The dilution of ambient air with humidified nitrogen was accounted for with a dilution factor of 1.3.

The 5 min detection limits (LOD), derived from the 2σ standard deviation of background (C2) measurements, ranged between 4.0 and 7.0 $s^{-1}$ (depending on PTR–QMS performance). Total uncertainty (1 σ) of the measurements was 35 % (median), with a precision of 16−41 % over 5 min depending on the quantity of reactivity.

## 2.6 Calculated OH reactivity from individually measured compounds (speciated OH reactivity)

Table S1 lists the 83 VOCs and 3 inorganic trace gases and their reaction rate constants considered for calculating speciated OH reactivity. The speciated OH reactivity is the sum of the OH reactivities attributed to individual (measured) trace gases (Eq. 2):



$R = \Sigma \, k_{VOC,i+OH} \, [VOC]_i$                                                                            (2)

Contributions of VOCs and inorganic trace gases (where *[VOC]* is their respective concentration in molecules cm$^{-3}$) to OH

reactivity (*R* in s$^{-1}$) are calculated using the gas-phase reaction rate constants ($k_{VOC+OH}$ in cm$^3$ molecule$^{-1}$ s$^{-1}$) of the respective

compounds with the OH radical. The difference between the sum of individual trace gas contributions to OH reactivity and

measured total OH reactivity is termed "unattributed" or "missing" OH reactivity. Out of a total of 86 chemical species that

were considered for the calculation of speciated reactivity here, 13 VOCs (specified in Table S1), including the known most

important contributors to OH reactivity in the tropical forest, were calibrated with gas standards and therefore have low

uncertainties in their concentrations as well as in their reaction rate coefficients (5−15 %). A further 70 exact masses (specified

in Table S1) monitored by PTR−ToF−MS were attributed to molecular formulae, and their concentrations derived using a

theoretical approach (Lindinger and Jordan, 1998), which has an uncertainty of ca. 50 %. In cases where several chemical

structures could be attributed to the measured mass, an average of the known reaction rate coefficients with OH was used for

calculating the speciated OH reactivity. In the few cases where rate coefficients were unknown in literature, the rate coefficient

of a VOC with comparable functional groups was applied (see Table S1). Due to the occasional large differences in rate

coefficients between possible structures, the uncertainty of the reaction rate coefficients $k_{VOC+OH}$ is estimated to be 100 %. The

uncertainty of the resulting speciated (i.e. calculated) OH reactivity depends on the fraction of gas-standard calibrated

compounds in the ambient air at any given point of time and varied between 15 % and 65 %, with an average (± standard

deviation) of 36 % ± 9 %. The relative composition of the total monoterpenes and sesquiterpenes measured by PTR-ToF-MS

was derived from TD-GC-ToF-MS data as specified in Table S1.

Periods influenced strongly by biomass burning were identified using an index defined as 3* acetonitrile mixing ratio in ppb

+ black carbon mass in µg m$^{-3}$. If this index was > 1 in the dry season or > 0.75 in the wet and transition season, the respective

data point was defined as strongly biomass burning influenced, whereas other data points were categorized as low biomass

burning influenced. This index was tested against the pristine conditions index PRBCUCO from Pöhlker et al. (2018) for

March 2018. In this period, ~80 % of the data points were categorized in the same way by both indices.

## 3. Results and discussion

### 3.1 Vertical, diel and seasonal variation of total OH reactivity

#### 3.1.1 Total OH reactivity profiles and diel cycles

Profiles of total OH reactivity between 80 m and 320 m above ground level at the ATTO tower are shown for different seasons

and times of the day in Fig. 1. Generally, noontime OH reactivity was higher in the dry seasons than in the wet or transition

seasons, which is consistent with earlier studies at lower heights (Nölscher et al., 2016). The overall lowest noontime (11:00–





15:00 LT (local time)) OH reactivity average was observed with 23.2 s$^{-1}$ at 80 m during the wettest period (June 2019, Fig. S1), and the highest noontime average with 39.4 s$^{-1}$ during the driest period of measurements (September 2019).

Nighttime (00:00–5:00 local time) averages were lower, between 14.1 and 20.2 s$^{-1}$. Interestingly, the noontime–nighttime difference was with 12.4–22.5 s$^{-1}$ much larger in the dry season than in the wet season (6.7 s$^{-1}$). The nighttime values appear

to be relatively consistent over the different measurement periods, whereas the noontime values are subject to larger seasonal differences. This is probably related to larger temperature differences between day and night in the dry season (see Sect. 3.3.1). In all seasons, there was a negative vertical gradient of OH reactivity towards higher levels around noon. This is expected, because noon and early afternoon is the time when the vegetation reaches its emission maximum (Sarkar et al., 2020). Reactive VOCs emitted by the vegetation become oxidized to less reactive compounds while they are transported aloft (see Sect. 3.2).

The vertical OH reactivity gradient was strongest in the driest period observed (September 2019; noontime gradient: -0.026 s$^{-1}$ m$^{-1}$, see Fig. S3) with a maximum difference between 80 m and 320 m of 6.8 s$^{-1}$, and weakest in the wet season (March 2018; noontime gradient: -0.005 s$^{-1}$ m$^{-1}$) with a maximum difference between 80 m and 320 m of 1.3 s$^{-1}$. The dry season vertical gradient was here approximately a factor of 4 lower than the OH reactivity gradient observed directly above the rainforest canopy from 24 m up to 80 m in the dry season 2013 (Nölscher et al., 2016). This behavior was also observed in

aircraft measurements of isoprene and monoterpenes, where vertical gradients inside the boundary layer above the rainforest decreased with height (Karl et al., 2007; Kuhn et al., 2007). In the wet season, the gradients measured here were comparable to those seen directly above the canopy in the 2013 wet season (Nölscher et al., 2016).

Between 80 and 320 m, the variability in the data (shown as standard deviations) was often as large as the gradient (Fig. 1). This suggests that turbulent mixing above the forest was often strong enough to transport reactive VOCs to higher levels faster

than the majority of the reactants could become oxidized to less reactive species. This observation reflects the well-mixed nature of air above rainforest canopies at daytime (Freire et al., 2017; Fuentes et al., 2016).

At nighttime, vertical gradients in total OH reactivity were barely discernable. The data suggest a slightly lower (by up to 1.8 s$^{-1}$) OH reactivity at the lowest observation level (80 m) than at 150 m and/or 320 m in all observation periods except September. The OH radical, the dominant VOC sink, is not available during the night to any significant degree, although some

can be generated by the reaction of ozone with terpenoids (Pfeiffer et al., 1998; Aschmann et al., 2002; Taraborrelli et al., 2012). Potentially, deposition to the surfaces of the forest canopy or to the soil becomes a relevant VOC sink visible in the 80 m values at night, while surfaces for dry deposition are less available at higher altitudes. VOC deposition on ecosystems, plants and even uptake to the cuticles has been observed in the past (Niinemets et al., 2014; Karl et al., 2010; Park et al., 2013; Langford et al., 2010; Cappellin et al., 2017). Soils can act as an additional VOC sink (Tang et al., 2019; Pegoraro et al., 2005;

Cleveland and Yavitt, 1997). Deposition might also partly explain the weaker vertical gradients in the wet season, when the overcast conditions led to weaker irradiation and therewith lower OH levels, while deposition continued or was even increased due to precipitation.





Total OH reactivity diel cycles (Fig. 2) reflect the typical diurnal behaviour of BVOC concentrations previously measured in
the rainforest (Langford et al., 2010; Yáñez-Serrano et al., 2015; Rizzo et al., 2010), which is broadened towards the evening
hours in comparison to emission fluxes (Sarkar et al., 2020; Kuhn et al., 2007). Sunrise was around 06:00 local time (LT) and
sunset around 18:00, independent of the season. In the mornings, OH reactivity started to increase in the 1–2 h following
sunrise, in parallel with temperature. The OH reactivity maximum occurred between 12:00 and 15:00 LT. At the end of the
day, OH reactivity decreased with PAR (photosynthetically active radiation), but, following the pattern of temperature more
closely than that of PAR, did not reach nighttime levels until 2–4 h after sunset.

Minimum total OH reactivity values were generally observed in the early morning before sunrise, coinciding with lowest diel
temperatures. The dependence of OH reactivity on temperature is discussed in more detail in Sect. 3.3.1.

In both dry seasons and the wet season, a second peak of OH reactivity, smaller than the noontime maximum, occurred during
sunset. Potentially, this was related a combination of a transport effect and terpenoid emissions spread out until the evening
(Pfannerstill et al., 2018).

The differences of OH reactivity diel cycles between heights were often within the standard deviation of diel hourly averages,
usually with slightly higher values at 80 m during the day as discussed above.

### 3.1.2 Total OH reactivity seasonality

Lowest temperatures and PAR values were observed during the wet season (March; hourly diel maxima 27.4 °C and
1166 µmol m$^{-2}$ s$^{-1}$), followed by the wet–dry transition season (June: 31.5 °C, 1955 µmol m$^{-2}$ s$^{-1}$), and the two dry season
periods (October: 33.1 °C, 1987 µmol m$^{-2}$ s$^{-1}$; September; 30.6 °C, 2230 µmol m$^{-2}$ s$^{-1}$). Surprisingly, we observed the lowest
daytime OH reactivity in the wet-dry transition season (June) with a diel hourly maximum at 80 m of 24.0 s$^{-1}$ when temperature
and irradiation were higher than in March (28.2 s$^{-1}$). However, as mentioned previously, the June 2019 transition season was
particularly wet with frequent rainfall and with a higher soil moisture than March 2018 (Fig. S1). These conditions appear to
have influenced BVOC concentrations, e.g. due to root inundation (Bracho-Nunez et al., 2012) or by a potential delay of leaf
flushing that usually starts in many trees during the wet-to-dry transition season (Lopes et al., 2016), which would have
influenced leaf phenology, another driver of tropical BVOC emissions (Alves et al., 2018; Wei et al., 2018). Also, wet
deposition was enhanced in June, contributing to reduced OVOC concentrations (see Sect. 3.2). Influence of rainfall on OH
reactivity is further discussed in Sect. 3.3.3. During the dry seasons, the diel hourly maxima of OH reactivity were 35.9 s$^{-1}$ and
41.5 s$^{-1}$ for October 2018 and September 2019, respectively. The larger daytime OH reactivity in September was accompanied
by higher PAR than in October.

In contrast to the daytime values, the nighttime OH reactivity in September 2019 was significantly lower than in October 2018.
One potential reason is found in enhanced nighttime ozone levels in September 2019, when nighttime average ozone mixing
ratios were 39 % (4.3 ± 1.7 ppb) higher than in October 2018 (Fig. 3). Ozone is a direct sink for terpenoid compounds, but
also generates secondary OH from the reaction with terpenes and sesquiterpenes (Pfeiffer et al., 1998; Aschmann et al., 2002;



Taraborrelli et al., 2012). This might have led to a larger nighttime VOC loss in September 2019. However, other potential explanations such as inter-annual differences in nighttime deposition cannot be ruled out.

The seasonal behavior in total OH reactivity observed here, with a maximum in September and a minimum in June, is consistent with satellite-derived isoprene observations from 2010–2011 (Alves et al., 2016). A comparison of the values presented here with OH reactivity measurements made in 2013 at the same site (at 79 m a. g. l. at the walk–up tower; Nölscher et al., 2016) shows very similar values between June 2013 and June 2019. In March 2013, the observed values were lower and in September 2013 they were higher than in the same months in 2018–2019. A potential reason for these differences could be

inter–annual variability. For example, March 2018 was less humid than March 2013 with median soil moistures of 0.169 $m^3$ $m^{-3}$ vs. 0.233 $m^3$ $m^{-3}$, and was relatively strongly influenced by biomass burning (Sect. 3.3.4). As discussed above, OH reactivity is generally higher under drier rainforest conditions (Fig. S1). Other factors not monitored, such as herbivore activity, can vary strongly between years (Velasque and Del-Claro, 2016), and can elicit the release of certain BVOCs that might contribute to inter-annual differences. Additionally, the VOC composition measured at the same height can differ

between the walk-up tower used in 2013 and the tall tower used in 2018–2019 due to local emissions (Zannoni et al., 2020a). The walk-up tower is built directly adjacent to trees, which might increase the influence and amount of very local, momentary BVOC emissions that could have led to the higher OH reactivities in September 2013, while the tall tower has a much larger footprint (Pöhlker et al., 2019) and is built on a small clearing.

**3.2 OH reactivity budget**

**3.2.1 OH reactivity speciation overview and unattributed reactivity**

In forested environments, the total OH sink is generally less well understood than in urban air masses (Yang et al., 2016; Williams and Brune, 2015; Zannoni et al., 2017). Previous total OH reactivity observations in tropical forests revealed large unattributed OH reactivity fractions of 50–79 % (Nölscher et al., 2016; Edwards et al., 2013; Sinha et al., 2008). The

unattributed fraction in these past studies was largest in the dry season at night (around 80 %), while it was smallest in the wet season around noon (5–50 %) (Nölscher et al., 2016). However, the number of trace gases included in these earlier investigations was restricted to only 15 to 19 species due to technical limitations. As the unattributed fraction depends on the number and relevance of trace gases included, we investigate here how well we can explain the OH reactivity budget by measuring a larger number of VOCs by PTR–ToF–MS.

We used 83 VOCs (for a complete list see Table S1) monitored by PTR–ToF–MS to compare with measured total OH reactivity. The VOCs which were calibrated by a theoretical approach (see Sect. 2.6 and Table S1) contributed between 29 % and 39 % of total OH reactivity. The unattributed fraction resulting from the inclusion of both gas-standard calibrated VOCs and those calibrated by the theoretical approach was usually in the range of the measurement uncertainty (Fig. 4) and within





the between-day variability (shown as standard deviation). Applying a more comprehensive VOC measurement technique, we
can thus close the OH reactivity budget within the uncertainty. The unattributed fraction was independent of measurement
height (Fig. S2), therefore data from 80 m is shown in Fig. 4 as a representative example.

The PTR–ToF–MS measurements revealed that an unexpectedly large fraction of total OH reactivity above the rainforest was
due to oxygenated VOCs (OVOCs). Only five OVOC species (methanol, acetaldehyde, acetone, methyl ethyl ketone, sum of
methacrolein + methyl vinyl ketone + ISOPOOH) could be monitored in previous studies, amounting to 10–15 % of speciated
and 4–5 % of total OH reactivity (Nölscher et al., 2016). A model has subsequently attributed 19 % of Amazon OH reactivity
to OVOCs (Safieddine et al., 2017). In addition to the aforementioned gas-standard calibrated OVOC species, here we include
OVOCs calibrated by a theoretical approach. As a result, we observe a higher OVOC-attributed speciated OH reactivity
fraction of 34–46 % (22–40 % of the total OH reactivity), attributed to 45 OVOC species. These include reactive oxygenates
that were not included in earlier observations, e.g. 3-methyl-furan (an isoprene oxidation product; Atkinson et al., 1989; Jardine
et al., 2013), and monoterpene oxidation products with the chemical formula $C_9H_{14}O$ (identified as nopinone and sabinaketone
by TD-GC-ToF-MS). Our findings support modelling results of Taraborrelli et al. (2012) which predicted that half of the OH
reactivity associated with the presence of isoprene would be due to its oxidation products.

The smallest daytime average OVOC OH reactivity at 80 m a.g.l. (4.2 s$^{-1}$, 21 % of the total) was observed in the rainiest
observation period (June 2019, with an average of 4.4 mm of rain per day), in contrast to the highest in the dry season (October
2018, 0.7 mm of rain per day) with 8.8 s$^{-1}$ (31 % of the total). This difference can be explained by a lower OVOC formation
rate associated with lower irradiation, and a higher tendency for wet deposition of OVOCs associated with the higher
precipitation rates in June (Langford et al., 2010).

In this study, isoprene accounted for 38–46 % of calculated and 23–43 % of total OH reactivity, and was thereby the single
chemical species with the largest contribution to OH reactivity. However, this percentage is lower than the 70 % of calculated
OH reactivity found at the Z14 station in Central Amazonia (Karl et al., 2007) and the 54–70 % of calculated OH reactivity
inside the canopy at the ATTO site in 2013 (Nölscher et al., 2016). However, Nölscher et al. (2016) already noted that isoprene
accounted for only ~20 % of the *total* OH reactivity during dry-season afternoons.

Our PTR–ToF–MS measurements also revealed several directly emitted BVOC species which were not included in earlier
rainforest OH reactivity studies – e.g. the sum of sesquiterpenes, oxygenated (e.g. citronellol, thymol) and aromatic
monoterpenes (e.g. p-cymene, cymenene), and several green leaf volatiles (GLVs, e.g. hexenol, hexenyl acetate, hexanal),
some of which are highly reactive towards OH. In total, we found 31 BVOCs that probably are direct emissions according to
literature, which is a large number compared to previous rainforest OH reactivity studies, but a small number in comparison
with a study that found 264 different VOCs in the emissions of tropical trees directly measured at the leaf/bark level (Courtois
et al., 2009). However, the most reactive among the directly emitted BVOCs are not expected to live long enough to reach the
80 m and higher observation platforms.



The sum of GLVs contributed 5–8 % of total OH reactivity. Monoterpenes and sesquiterpenes contributed on average 2–7 % and 0.1–0.9 % of total OH reactivity, respectively. In summary, all groups of primary BVOCs except for isoprene were smaller OH sinks than the summed OVOCs. However, it has to be noted that some OVOCs can be primary plant emissions and do not

necessarily have to be the result of atmospheric oxidation processes (Harley et al., 2007; Schade and Goldstein, 2002; Rottenberger et al., 2004; Bracho-Nunez et al., 2013; Niinemets et al., 2014). Even isoprene oxidation products can be directly emitted to a certain extent (Jardine et al., 2013).

The OH reactivity of CO and methane (grouped as "inorganics", Fig. 4) amounted to 3–4 % of the total, with CO mainly varying under biomass burning influence. The OH reactivity attributed to aromatics, sulfur- and nitrogen-containing VOCs

(grouped under "others") was small with 0–1% of the total.

The three individual species that contributed most to daytime OH reactivity were in both dry seasons and in the transition season isoprene > sum of methacrolein (MACR) + methyl vinyl ketone (MVK) + ISOPOOH > monoterpenes. In the wet season they were isoprene > sum of MACR + MVK + ISOPOOH > sum of hexenol + hexanal.

Previous studies explained unattributed OH reactivity fractions in the rainforest with a mixture of unmeasured primary BVOCs (e.g. green leaf volatiles, sesquiterpenes) and BVOC photooxidation products (Nölscher et al., 2016), or with oxidation products alone (Edwards et al., 2013). Edwards et al. (2013) created a box model to account for unattributed OH reactivity in a South-East Asian rainforest, which included ~ 900 oxidized intermediates that contributed 47 % of the calculated OH reactivity. Our data, with 23–43 % of total OH reactivity attributed to 45 OVOC species and ~10 % to several previously

unaccounted for primary BVOCs (including green leaf volatiles, oxygenated and aromatic monoterpenes, and sesquiterpenes), indicate that previously unattributed OH reactivity fractions were due to both primary BVOCs and oxidation products, with oxidation products dominating.

Assuming that the remaining unattributed OH reactivity fraction here is significant despite being within the measurement uncertainty (Fig. 4), 7–35 % of the total OH reactivity (seasonal averages) remain unexplained here. This fraction was lowest

in the dry season afternoons (4–11 %) – interestingly, the time of the day with largest unattributed OH reactivity in Nölscher et al. (2016). This is also the time with largest OVOC reactivity, which explains why closure could be significantly improved for the afternoons here compared to previous studies.

Over all seasons, the unattributed fraction was largest and most variable in the early morning (Fig. 4, 28–39 %). This is when vegetation starts its diurnal photosynthetic activity with the largest $CO_2$ uptake and stomatal conductance of the day

(Pfannerstill et al., 2018). At this time of the day, emission of primary BVOCs from storage pools and from build-up within the leaf while the stomata were closed is likely, suggesting that part of the remaining unaccounted for OH reactivity in this work could be due to direct plant emissions. The number of different VOCs emitted by vegetation is potentially much larger than the number we were able to measure (Courtois et al., 2009). A peak in sesquiterpene mixing ratios co-occurs with the early-morning peak in unattributed OH reactivity (Zannoni et al., 2020b), perhaps indicating a common source of the

sesquiterpenes and the unidentified reactants.





As the PTR–ToF–MS is not able to separate or identify all OVOC species and misses unstable intermediates, further unmeasured secondary oxygenates could be responsible for the remaining unattributed OH reactivity in this study. The fact that the unattributed OH reactivity fraction appears to be independent of altitude (Fig. S2) may be an indicator for long-lived OVOC species contributing to the remaining unattributed fraction. However, it has to be noted again that the uncertainty of
the measurement was generally larger here than the unattributed fraction.

### 3.2.3 Vertical and diel trends in OH reactivity speciation

OH reactivity speciation at the ATTO tower reveals height-related trends (Fig. 5 (a-b), Table 1).
In all seasons during daytime, the OH reactivity contribution of directly emitted BVOCs (isoprene, terpenoids) decreased with height (e.g. for isoprene from 51 % at 80 m to 39 % at 320 m in October 2018), while the fraction of oxidation products, i.e. OVOCs, increased (e.g. from 31 % to 39 % in October 2018). This is due to the increasing photochemical age of the air mass with height, which increases the abundance of OVOCs relative to primary BVOCs such as isoprene (Karl et al., 2009).
In order to compare the relevance of the different VOC classes for the carbon cycle with their relevance as OH sinks, Fig. 5
(c-d) shows carbon concentration. If the carbon concentration is considered instead of the OH reactivity, the fraction of OVOCs (daytime: 52–66 %) dominates over the isoprene fraction (daytime: 12–27 %) at every height. The intense solar irradiation, high temperature and humidity in the tropical rainforest environment lead to rapid conversion of primary BVOCs to secondary oxygenated compounds (Karl et al., 2009). Our results indicate that a considerable part of the BVOCs directly emitted by vegetation is already oxidized when reaching 80 m, but since oxidation of terpenoids decreases their reactivity (Nölscher et
al., 2014), their contribution to the total OH sink is smaller than their contribution to total carbon concentration (Fig. 5 c-d) or their contribution to total VOC mixing ratio (Fig. 5 e-f).

At nighttime, the composition of OH reactivity was shifted towards OVOCs. They accounted for 23–40 % of the total nighttime OH reactivity, while isoprene contribution decreased to 19–36 %. This reflects that OVOCs produced during the day have
longer lifetimes than isoprene, and therefore are not as efficiently removed from the atmosphere before nightfall. In terms of carbon, the OVOC fraction increases to 57–73 % at night.
The positive nighttime vertical gradient towards higher altitudes which is recognizable in total OH reactivity (see Sect. 3.1.1) is more distinct in carbon concentration and mixing ratios (Fig. 5 d, f). This supports the conclusion that deposition to the canopy may become a relevant VOC sink at night (see Sect. 3.1.1). The compound group predominantly responsible for the
positive upwards gradient is OVOCs with a 16 % larger nighttime OVOC reactivity and 23 % larger OVOC mixing ratio at 320 m compared to 80 m (October 2018). With their high polarity and molecular mass, OVOCs are likely to be subject to dry deposition to surfaces.





### 3.3 Dependence of OH reactivity on environmental parameters

Total OH reactivity observed above the rainforest can change due to variation in several environmental factors, e.g. the release of BVOCs from storage compartments or after in-situ de-novo production from vegetation, transport processes to the point of observation, other meteorological factors such as wind and rain, or biomass burning. In this section we investigate how total OH reactivity changed with these factors, in the hope of gaining insights that can be used in modelling Amazon forest OH reactivity.

### 3.3.1 Temperature


Diel cycles (Sect. 3.1) showed that the patterns of total OH reactivity and ambient temperature were similar. Plant BVOC emissions are known to depend on temperature and are thought to provide, among other advantages, a heat stress coping mechanism (e.g. Penuelas and Munne-Bosch, 2005; Sharkey et al., 2008). Since total OH reactivity above the rainforest is dominated by primary BVOCs (Sect. 3.2), we investigate here the dependence of total OH reactivity measured at 80 m a. g. l.
on temperature measured at canopy height, i.e. at 26 m a. g. l. The diel variation of temperature between the night and the warmest time of the day (~15:00–17:00 LT) amounted to up to $\approx$ 8 °C in the wet season, 10 °C in the transition season and 12 °C in the dry season (Fig. 5). Seasonal larger day-night temperature differences were accompanied by larger day-night OH reactivity differences (see Sect. 3.1.1). Figure 6 shows total OH reactivity as a function of temperature, which can be parameterized using an exponential function (Table 2), adapting the procedure used for the parameterization of temperature-
dependent BVOC emissions in the MEGAN model (Guenther et al., 2012):

$$R = R_S \exp(\beta[T - T_S]) \tag{3},$$

where $R$ is total OH reactivity, $T$ is the temperature, $T_s$ is the standard temperature (25 °C), $R_s$ is the total OH reactivity at standard temperature, and $\beta$ is an empirical coefficient.

The dependence of OH reactivity on temperature appears to be strongest in the dry season. The parameterization for both dry
seasons (Fig. 6 b, d) was unified with a common exponential fit equation as displayed in Table 2.

The color scaling in Fig. 6 shows that higher temperature and OH reactivity often co-occurred with higher PAR because temperature is driven by PAR at daytime. PAR is an additional driver of reactive emissions in the rainforest (Kuhn et al., 2004a; Jardine et al., 2015). However, there is a PAR-independent temperature dependence visible at PAR = 0, i.e. during the night (Fig. 6), which is why we chose to parameterize OH reactivity based on temperature rather than PAR. Air temperature
can serve as a proxy for the combined effects of direct light- and temperature-dependent emission as well as transport, which all influence observed total OH reactivity. As temperature at canopy height correlated well with OH reactivity measured at 80 m, the results discussed here indicate that plant response to environmental variables is still visible at 80 m, i.e. ~ 50 m above the canopy top.





The seasonal differences between the temperature-related parameterization functions imply that a temperature dependence alone cannot explain OH reactivity. Additional drivers of isoprenoid emissions such as leaf phenology, biotic stress, or root inundation (potentially relevant in June 2019, see Sect. 3.1.2) may be contributing factors in seasonal differences of dependence on temperature (Alves et al., 2018; Kuhn et al., 2004b; Bracho-Nunez et al., 2012). Also, differences in turbulent transport, precipitation and biomass burning influence the observed OH reactivity (see following sections). Nevertheless, total OH reactivity modeled from the temperature-dependent parameterization is a reasonable estimate of the measured values (Fig. S4, $r^2 = 0.61$), although the deviations indicate that additional stochastic drivers may need to be considered to improve the reproduction of observed total OH reactivity.

### 3.3.2 Wind speed/turbulence

The wind speed observed at 26 m (inside the canopy) was on average $0.37 \pm 0.33$ m s$^{-1}$ during the observation periods. In order to investigate the influence of wind speed on OH reactivity, data were categorized according to wind speed and day/night with high inside-canopy wind speed > 0.4 m s$^{-1}$ and low inside-canopy wind speed < 0.2 m s$^{-1}$. Daytime was defined as PAR > 0. Figure 7 (a) shows that periods of high wind speed were associated with higher OH reactivity in all seasons, with more pronounced differences at daytime. The daytime difference of median OH reactivity between periods with wind speed > 0.4 m s$^{-1}$ and < 0.2 m s$^{-1}$ was 4.0 s$^{-1}$, 7.3 s$^{-1}$, 5.4 s$^{-1}$, and 15.0 s$^{-1}$ for March 2018, October 2018, June 2019, and September 2019, respectively. The nighttime differences were not as strong or even opposite with -1.1 to 0.7 s$^{-1}$.

Several studies showed that leaf damage through storms can lead to strong emissions of green-leaf volatiles (GLVs) and terpenes (Bouvier-Brown et al., 2009; Maleknia et al., 2009; Bamberger et al., 2011; Ruuskanen et al., 2011; Kaser et al., 2013). If increased leaf damage at higher wind speeds was the cause for the differences in OH reactivity here, GLVs (Fig. 7 (b)), i.e. indicators for leaf damage, would exhibit a strong wind-dependent pattern. This is not the case. In some observation periods, e.g. October 2018, it is even opposite (i.e. higher GLV at low wind speed). Additionally, the effect does not appear to be prevalent at night. Therefore, we conclude that the observed correlation of higher atmospheric reactivity with higher wind speed cannot be attributed to leaf damage, but to other effects. As higher wind speed correlates with higher temperature during daytime and OH reactivity strongly correlates with temperature (Sect. 3.3.1), we cannot differentiate the causes for the observed wind–reactivity correlations with certainty. One possible wind-related effect is more efficient transport to 80 m at higher wind speeds. This means that the reactants emitted by vegetation would be transported to higher altitudes faster in such cases and consequently remain non-oxidized (and more reactive) up to higher altitudes. This effect is larger during daytime because the abundance of highly reactive compounds in the canopy that are available for upward transport is larger, and because there is stronger mixing than at night (Freire et al., 2017; Fuentes et al., 2016). It has been shown previously that conditions for vertical transport are more favorable under drier and warmer conditions, leading to larger OH reactivity observed





above the canopy (Pfannerstill et al., 2018). This is supported by our data that indicate a larger difference between high and low wind speed reactivity in the dry season than in the wet or transition season.

### 3.3.3 Precipitation

Amazonian rain events are usually due to convective storms and occur most frequently between 12:00 and 17:00 LT (Tanaka et al., 2014; Oliveira et al., 2020). The influence of rain on total OH reactivity was investigated using data from June 2019, a period with almost daily precipitation events. They were associated with sharp temperature drops due to convective cooling (Fig. 8). The time series in Fig. 8 (a) shows an increase in hourly OH reactivity during rain events and a steep and distinct drop in OH reactivity after rain events. The composition of OH reactivity during rain events and dry periods was similar. However, the unattributed fraction was larger ($39 \pm 19$ % vs. $23 \pm 25$ %) during rain events.

The increase of the hourly average *during* the rain events was surprising given the large contribution of OVOCs to total OH reactivity and their high water solubility. The short-term spike in observed OH reactivity during rain events can be attributed to both re-volatilization of VOCs from surfaces after wetting, and the co-occurring peak in wind speed (Fig. 8), which indicates turbulent mixing, associated with the precipitation events (Sect 3.3.2, Oliveira et al., 2020). Similarly, rapid vertical transport of particles has been observed during Amazon rain events (Wang et al., 2016).

The OH reactivity spikes were highly variable (increasing OH reactivity by $7.7 \pm 6.5$ s$^{-1}$ or $40 \pm 34$ %) and coincided with peak mixing ratios of isoprene and monoterpenes (Fig. 8 d), indicating wind–facilitated upward transport from the canopy to the point of observation. The large variability is due to the fact that the size of the increase in OH reactivity depends on the trace gas concentrations available for turbulent transport, and therefore on the time of the day.

As an additional potential influence of rain events, there is evidence of increased BVOC emissions from wetted plant surfaces of pine trees (Kim, 2001; Janson, 1993). This may contribute to the OH reactivity spike during the rain events observed here. Soil and litter have also been shown to emit bursts of VOCs during and after rain events (Greenberg et al., 2012; Bourtsoukidis et al., 2018; Rossabi et al., 2018), attributed to both humidity-dependent microbial activity and physical desorption of compounds on the soil surfaces (the Birch effect). However, with the large distance between the point of observation and the soil in our study, we do not expect a large effect of soil emissions on the OH reactivity at 80 m above ground level. Convective mixing during daytime storm events did not affect stable stratification in the Amazon forest canopy in a study presented by Oliveira et al. (2020). Stable thermal stratification close to the ground can stop soil emissions from reaching higher altitudes in the rainforest (Kruijt et al., 2000; Gerken et al., 2017; Santana et al., 2018; Pfannerstill et al., 2018), and it has been shown that ozonolysis inside the canopy already strongly decreases reactant levels before they reach the canopy top (Jardine et al., 2011; Jardine et al., 2015; Bourtsoukidis et al., 2018). Fig. 8 (c) shows that a rain event occurring in the early morning barely increased OH reactivity, in contrast to a rain event at noon on the same day. This indicates that the reactants mainly involved





in the rain reactivity peaks are subject to a diurnal cycle, as plant emissions are. Nevertheless, it cannot be ruled out that part

of the additional unattributed reactivity during the precipitation events might be due to soil emissions.

The reduction in OH reactivity observed *after* each rain event can be explained by the co-occurring drop in temperature and irradiation, which both strongly influence plant emissions and therewith observed OH reactivity (see Sect. 3.3.1). The time lag between the temperature- and PAR-induced decrease in plant emissions and reduced OH reactivity at the tower after the rain

event may be due to a time lag in plant reactions and transport times. Additionally, water soluble VOCs, especially OVOCs, can be removed from ambient air by wet deposition on the scale of the rain-affected area (Langford et al., 2010; Warneck and Williams, 2012). Figure 8 (e) indicates such a decrease in OVOC-attributed reactivity after the rain event in the order of $\approx 30$ % compared to the rain-preceding values. The vertical distribution of OVOCs before, during and after two example rain events is shown in Fig. S5. There is no indication of re-volatilization from surfaces playing a role in OVOC OH reactivity after the

rain event, as the values measured at 80 m do not surpass those at the higher altitudes except for in the turbulence-related spike during precipitation. It has to be noted that an effect of re-volatilization would be difficult to separate from influences of turbulence effects and from the recovery of photochemical processes after precipitation.

We quantified the after–rain effect on total OH reactivity values by subtracting the OH reactivity in the data point following the rain event (hourly average) from the average of the OH reactivity preceding the rain event and 2 h after. The reduction in

OH reactivity after rain events was $2.5 \pm 5.2$ s$^{-1}$ or $13 \pm 23$ % (average $\pm$ standard deviation) for 14 rain events in June 2019. As shown in Fig. 8 (b)–(e), the recovery to the typical diel cycle or pre-rain values after each precipitation event took up to 3– 4 h for total OH reactivity, and 12 h for OVOCs, which suggests an overall negative effect of rainfall on total OH reactivity above the rainforest despite the short-term increase observed during the precipitation events.

### 3.3.4 Biomass burning

Air masses influenced by forest fires regularly reach the ATTO site (Pöhlker et al., 2019; Pöhlker et al., 2018). Biomass burning influenced air masses are characterized by several markers, e.g. black carbon (Saturno et al., 2018), benzene and acetonitrile (Yáñez-Serrano et al., 2015). Using an index derived from black carbon mass concentration and acetonitrile mixing ratios, OH reactivity observations were divided into time points strongly influenced by biomass burning and time points

without or with low biomass burning influence (Sect. 2.6).

The results, displayed in Fig. 9, show that stronger biomass burning influence increases the OH reactivity observed at the ATTO tower. Given the fact that biomass burning releases many different trace gases (e.g. Andreae and Merlet, 2001; Koss et al., 2018), this finding is not unexpected. The difference between biomass burning and low or no biomass burning reactivity was on average $9.5 \pm 2.5$ s$^{-1}$ ($22 \pm 5$ %) during daytime, and $2.7 \pm 0.8$ s$^{-1}$ ($10 \pm 3$ %) at night. This day-night difference in the

biomass burning effect on total OH reactivity may be due to a) local manmade biomass burning usually started during the day and b) the fact that aromatic compounds emitted by such fires have low OH reactivities, while their photooxidation products,





which only can be formed during the day, have higher OH reactivities (e.g. by a factor of > 10 between benzene and its oxidation product phenol).

The effect of biomass burning on OH reactivity is smaller than the 20–50% VOC mixing ratio increase due to Amazonian
pasture fires reported by Christian et al. (2007), because compounds emitted by forest fires (e.g. aromatics, acetonitrile; Koss et al., 2018) are relatively unreactive towards the OH radical compared to e.g. the terpenoids emitted by plants. For example, the reaction rate coefficient of acetonitrile is 4 orders of magnitude lower than that of isoprene. The dependence of total OH reactivity at ATTO on the activity of vegetation, characterized by isoprene abundance, is therefore stronger than its dependence on biomass burning (Fig. S6). This is also reflected in the large variability in both the biomass burning and low biomass
burning data (Fig. 9), i.e. the categories are not particularly distinct in total OH reactivity. In contrast to Kumar et al. (2018), who found larger unattributed OH reactivity fractions in biomass burning plumes than in non-biomass burning ambient air over the Indo-Gangetic Plain, we found no significant difference of unattributed fractions between biomass burning and no or low biomass burning air at the ATTO site, because here the effect of direct biogenic emissions dominates over that of biomass burning.

As the probability of drought increases in Amazonia with climate change and land use change (Swann et al., 2015; Baidya Roy, 2002; Malhi et al., 2008; Staal et al., 2020), so does the probability of forest fires (Fonseca et al., 2019). The here observed effects of biomass burning on OH lifetime and, consequently, atmospheric residence times of greenhouse gases and pollutants above the Amazon forest, may therefore become larger and/or more relevant in the future.

**4. Summary and conclusions**

The first measurements of total OH reactivity above the Amazon rainforest between 80 m and 320 m a. g. l. were conducted during four campaigns covering three different seasons: A wet season (March 2018), a wet–dry transition season (June 2019), and two dry seasons (October 2018, September 2019). A typical BVOC-related seasonality with lower OH reactivity in the wet and transition seasons (averages of 23.7 s$^{-1}$ and 19.9 s$^{-1}$ at daytime in March and June, respectively) and highest values in
the dry season (averages of 29.1 s$^{-1}$ and 28.2 s$^{-1}$ at daytime in September and October, respectively) was observed.

Daytime vertical gradients of the OH sink above 80 m, decreasing towards higher altitudes, were about a factor of 4 lower than gradients between the canopy top and 80 m, indicating well-mixed conditions that allow transport of reactive VOCs from 80 m to higher levels faster than the majority of the reactants could become oxidized to less reactive species. This was visible in a
vertical trend of OH reactivity composition with decreasing primary BVOCs and increasing secondary OVOCs towards higher levels. A wind speed dependency of OH reactivity measured at 80 m supports the conclusion of a strong impact of turbulent transport on the OH reactant levels observed above the canopy.



At nighttime, the vertical gradient was inversed, i.e. total OH reactivity (and even more so VOC mixing ratios) increased towards higher altitudes. This feature was attributed to deposition to the soil or canopy becoming a relevant VOC sink during
the night.

The diel cycle of OH reactivity followed temperature patterns with the diel maximum around noon or in the early afternoon. Potentially due to larger day–night temperature differences, the day–night total OH reactivity difference was larger in the dry seasons (5.9–11.3 s$^{-1}$) than in the wet season (3.7 s$^{-1}$) or in the transition season (4.9 s$^{-1}$). The lowest nighttime total OH reactivity of all observation periods was measured in September 2019, which could potentially be related to elevated ozone
levels that were available for nighttime VOC oxidation.

We investigated the influence of environmental factors on OH reactivity, showing that total OH reactivity above the rainforest increased shortly during rain events due to convective transport. However, precipitation had an overall decreasing effect on total OH reactivity because of the drop in temperature and irradiation associated with these events, and because of the wet
deposition of OVOCs. After rain events, the OVOC-attributed OH reactivity decreased by ≈ 30 %. The OH reactivity of OVOCs was smallest in the observation period with most rain, likely due to wet deposition.

Biomass burning increased OH reactivity by 2.7 s$^{-1}$ at night and 9.5 s$^{-1}$ during the day (10–22 %).

The temperature dependence of total OH reactivity was parameterized for dry, transition and wet seasons. Total OH reactivity modelled from this parameterization reproduced measured values reasonably well. This parameterization could help to
simplify and validate models of Amazon rainforest OH chemistry, because total OH reactivity is one holistic parameter that directly defines the OH loss frequency. Total OH reactivity could thus be used in simulations instead of many different BVOCs, where part of the OH sink is likely to be missed.

Studies of total OH reactivity in forested environments often showed significant "missing", i.e. unattributed fractions of total
OH reactivity. In an attempt to close the OH reactivity budget, we applied a more comprehensive VOC detection technique than was used in previous Amazon forest studies. The OH reactivity of compounds with uncertain structure from PTR−ToF−MS measurements was calculated using averages of the reaction rate constants of all possible structures attributed to each chemical formula. These species were important OH sinks, accounting for 29−39 % of the seasonal total OH reactivity. Using 83 VOCs and 3 inorganic trace gases, the budget of total OH reactivity was closed within the measurement uncertainty,
suggesting that previously unaccounted for OH reactivity in the rainforest was due to a combination of both primary BVOCs and oxygenated compounds. Our analysis reveals an important contribution of OVOCs to seasonal total OH reactivity of 22–40 %, which is a factor of > 5 more than detected in past rainforest studies, and a factor of ≈ 2 more than predicted by an atmospheric chemistry model (Safieddine et al., 2017). An explanation for the large contribution of OVOCs to the OH sink is found in the meteorological conditions in the tropics with intense solar irradiation, high humidity (and therefore high OH
levels), and elevated temperatures, causing fast photochemical oxidation processes. Additionally, some OVOCs can be directly emitted by plants. We also accounted for several primary BVOCs that were not included in earlier studies, e.g. green leaf

volatiles, sesquiterpenes, and aromatic monoterpenes. Isoprene, generally considered the main OH sink above the tropics, contributed less than 45 % of the total OH reactivity. This shows that comprehensive measurement techniques are necessary to understand the OH sink above tropical forests, and that the previously underestimated contribution of OVOCs needs to be
included in models of the tropical forest OH sink.

**Data availability**

The data used in this study will be available via the ATTO data portal at http://attodata.org/.

**Author contributions**

EP, NR, AE, and AR conducted OH reactivity measurements. AE and AR were responsible for VOC measurements and data.
EP analyzed the data and wrote the first draft of the manuscript. Ozone data and the inlet system were provided by SW, AT and MS; methane and carbon monoxide data by DW and JL; black carbon data by BH, DW, FD and CP; speciated terpene data by NZ, and meteorological data by MOS, AA, DW, FD and CP. JW supervised the study. All authors contributed to manuscript writing and revision, read and approved the submitted version.

**Competing interests**

The authors declare that they have no conflict of interest.

**Special issue statement**

This article is part of the special issue "Amazon Tall Tower Observatory (ATTO)".

**Acknowledgements**

The authors would like to express their gratitude to the ATTO team (particularly Reiner Ditz and Hermes Braga Xavier) for
technical and logistical support, and to Antonio Huxley Melo Nascimento and Thomas Klüpfel for help with the PTR–MS measurements. We thank Rodrigo Souza for providing an ozone monitor and Mohammad Kazziha for GIS plots. We are grateful for continuous support in establishing the ATTO tower by the Max Planck Society and the Instituto Nacional de Pesquisas da Amazônia. We acknowledge the funding by the German Federal Ministry of Education and Research (BMBF contracts 01LB1001A, 01LK1602A and 01LK1602B) and the Brazilian Ministério da Ciência, Tecnologia e Inovações
(MCTI/FINEP contract 01.11.01248.00) as well as the Amazon State University (UEA), FAPESP, CNPq, FAPEAM,





LBA/INPA, and SDS/CEUC/RDS-Uatumã. NZ was supported by the European Commission Horizon 2020 (grant no. FETOPEN-737071) ULTRACHIRAL Project.

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





**Table 1: Overview of average measured total and speciated OH reactivity and unattributed (missing) fraction of total OH reactivity by season, measurement height and separated by day and night. Values shown are averages ± standard deviations. The chemical species attributed to each group are listed in Table S1. For an explanation of unattributed or speciated OH reactivity, refer to Sect. 2.5. MT: monoterpenes, SQT: sesquiterpenes, GLV: green leaf volatiles, OVOCs: oxygenated VOCs.**

| | | OH reactivity (s⁻¹) of | | | | | | | Unattributed fraction | Measured total OH reactivity (s⁻¹) |
| | | Isoprene | MT | SQT | GLV | OVOCs | Other VOCs | Inorganic trace gases | | |
|---|---|---|---|---|---|---|---|---|---|---|
| **March 2018** | | | | | | | | | | |
| Daytime | 80 m | 8.69 ± 3.90 | 1.62 ± 0.29 | 0.16 ± 0.02 | 1.11 ± 0.15 | 5.56 ± 0.70 | 0.15 ± 0.02 | 0.66 ± 0.02 | 24 ± 19 % | **23.70 ± 6.52** |
| Daytime | 150 m | 7.65 ± 3.64 | 1.56 ± 0.27 | 0.12 ± 0.02 | 1.12 ± 0.17 | 5.61 ± 0.62 | 0.15 ± 0.02 | 0.62 ± 0.02 | 28 ± 21 % | **23.51 ± 6.86** |
| Daytime | 320 m | 6.33 ± 2.93 | 1.47 ± 0.25 | 0.11 ± 0.01 | 1.12 ± 0.17 | 5.92 ± 0.68 | 0.15 ± 0.02 | 0.56 ± 0.02 | 26 ± 21 % | **21.56 ± 6.18** |
| Nighttime | 80 m | 5.42 ± 1.93 | 1.43 ± 0.17 | 0.14 ± 0.02 | 1.12 ± 0.06 | 5.55 ± 0.88 | 0.15 ± 0.03 | 0.66 ± 0.01 | 27 ± 23 % | **20.02 ± 4.64** |
| Nighttime | 150 m | 6.13 ± 2.12 | 1.42 ± 0.13 | 0.11 ± 0.01 | 1.12 ± 0.06 | 6.89 ± 0.59 | 0.15 ± 0.03 | 0.62 ± 0.00 | 23 ± 24 % | **20.10 ± 3.40** |
| Nighttime | 320 m | 5.92 ± 2.10 | 1.36 ± 0.13 | 0.10 ± 0.01 | 1.12 ± 0.06 | 6.26 ± 0.55 | 0.16 ± 0.03 | 0.56 ± 0.00 | 23 ± 23 % | **20.00 ± 3.82** |
| **October 2018** | | | | | | | | | | |
| Daytime | 80 m | 14.27 ± 7.23 | 1.21 ± 0.58 | 0.09 ± 0.05 | 1.48 ± 0.50 | 8.78 ± 3.07 | 0.75 ± 0.12 | 0.80 ± 0.05 | 2 ± 25 % | **28.05 ± 7.92** |
| Daytime | 150 m | 12.51 ± 5.67 | 0.97 ± 0.44 | 0.06 ± 0.03 | 1.37 ± 0.38 | 9.13 ± 2.65 | 0.72 ± 0.10 | 0.69 ± 0.03 | 7 ± 22 % | **27.50 ± 6.71** |
| Daytime | 320 m | 9.98 ± 5.05 | 0.74 ± 0.39 | 0.04 ± 0.03 | 1.42 ± 0.38 | 10.06 ± 2.38 | 0.72 ± 0.10 | 0.63 ± 0.04 | 11 ± 26 % | **25.64 ± 6.04** |
| Nighttime | 80 m | 7.80 ± 5.28 | 0.58 ± 0.30 | 0.06 ± 0.04 | 1.45 ± 0.52 | 7.91 ± 3.52 | 0.72 ± 0.07 | 0.82 ± 0.07 | 13 ± 32 % | **22.20 ± 5.12** |
| Nighttime | 150 m | 7.26 ± 3.94 | 0.48 ± 0.22 | 0.04 ± 0.03 | 1.38 ± 0.40 | 8.59 ± 3.09 | 0.67 ± 0.07 | 0.72 ± 0.05 | 16 ± 27 % | **22.80 ± 5.21** |
| Nighttime | 320 m | 6.99 ± 3.92 | 0.46 ± 0.23 | 0.03 ± 0.02 | 1.38 ± 0.38 | 9.21 ± 3.09 | 0.65 ± 0.09 | 0.65 ± 0.05 | 15 ± 29 % | **22.94 ± 4.75** |
| **June 2019** | | | | | | | | | | |
| Daytime | 80 m | 6.92 ± 3.88 | 1.13 ± 0.75 | 0.16 ± 0.12 | 1.41 ± 0.72 | 4.17 ± 1.94 | 0.07 ± 0.05 | 0.62 ± 0.03 | 27 ± 27 % | **19.85 ± 6.15** |
| Daytime | 150 m | 5.59 ± 3.37 | 1.02 ± 0.77 | 0.14 ± 0.11 | 1.40 ± 0.79 | 4.17 ± 1.77 | 0.07 ± 0.05 | 0.59 ± 0.03 | 34 ± 28 % | **19.55 ± 5.50** |
| Daytime | 320 m | 4.78 ± 3.26 | 0.99 ± 0.77 | 0.14 ± 0.11 | 1.37 ± 0.70 | 4.51 ± 1.70 | 0.07 ± 0.05 | 0.52 ± 0.03 | 30 ± 32 % | **17.64 ± 5.48** |
| Nighttime | 80 m | 3.33 ± 1.81 | 0.83 ± 0.69 | 0.15 ± 0.12 | 1.26 ± 0.68 | 3.38 ± 1.47 | 0.06 ± 0.04 | 0.62 ± 0.02 | 35 ± 35 % | **14.93 ± 2.95** |
| Nighttime | 150 m | 3.07 ± 1.61 | 0.77 ± 0.67 | 0.13 ± 0.11 | 1.24 ± 0.66 | 3.72 ± 1.44 | 0.06 ± 0.04 | 0.58 ± 0.02 | 37 ± 36 % | **15.19 ± 2.16** |
| Nighttime | 320 m | 2.73 ± 1.61 | 0.74 ± 0.64 | 0.13 ± 0.10 | 1.23 ± 0.63 | 4.13 ± 1.38 | 0.06 ± 0.04 | 0.51 ± 0.02 | 33 ± 38 % | **14.32 ± 2.42** |
| **September 2019** | | | | | | | | | | |
| Daytime | 80 m | 13.22 ± 6.98 | 1.21 ± 0.46 | 0.09 ± 0.02 | 1.67 ± 0.41 | 7.85 ± 2.44 | 0.03 ± 0.01 | 0.79 ± 0.03 | 15 ± 22 % | **29.13 ± 10.78** |
| Daytime | 150 m | 10.69 ± 5.81 | 1.02 ± 0.41 | 0.08 ± 0.02 | 1.40 ± 0.36 | 7.14 ± 2.02 | 0.03 ± 0.00 | 0.75 ± 0.03 | 14 ± 29 % | **24.68 ± 9.80** |
| Daytime | 320 m | 8.80 ± 5.04 | 0.87 ± 0.37 | 0.07 ± 0.02 | 1.43 ± 0.51 | 7.17 ± 1.88 | 0.03 ± 0.01 | 0.68 ± 0.03 | 18 ± 31 % | **23.20 ± 9.59** |
| Nighttime | 80 m | 5.57 ± 3.37 | 0.67 ± 0.25 | 0.07 ± 0.02 | 1.43 ± 0.41 | 5.98 ± 1.85 | 0.03 ± 0.01 | 0.78 ± 0.05 | 17 ± 36 % | **17.85 ± 5.96** |
| Nighttime | 150 m | 6.12 ± 3.32 | 0.66 ± 0.22 | 0.07 ± 0.02 | 1.38 ± 0.37 | 6.27 ± 1.84 | 0.03 ± 0.01 | 0.74 ± 0.06 | 9 ± 41 % | **17.24 ± 5.45** |
| Nighttime | 320 m | 4.94 ± 2.68 | 0.57 ± 0.19 | 0.06 ± 0.02 | 1.35 ± 0.36 | 6.27 ± 1.44 | 0.03 ± 0.01 | 0.68 ± 0.06 | 9 ± 45 % | **15.67 ± 4.55** |



**Table 2. Parameterization of total OH reactivity in dependence of temperature: Coefficients for the equation $R = R_s \exp(\beta[T-T_s])$, with $R$ = total OH reactivity, $T$ = temperature in °C, $T_s$ = standard temperature (25 °C), $R_s$ = total OH reactivity at standard temperature, and $\beta$ is an empirical coefficient.**

| Season | $R_s$ | $\beta$ |
|---|---|---|
| **Dry** | $19.9 \pm 0.4$ | $0.076 \pm 0.004$ |
| **Wet** | $20.8 \pm 0.4$ | $0.066 \pm 0.008$ |
| **Transition** | $14.4 \pm 0.4$ | $0.051 \pm 0.005$ |




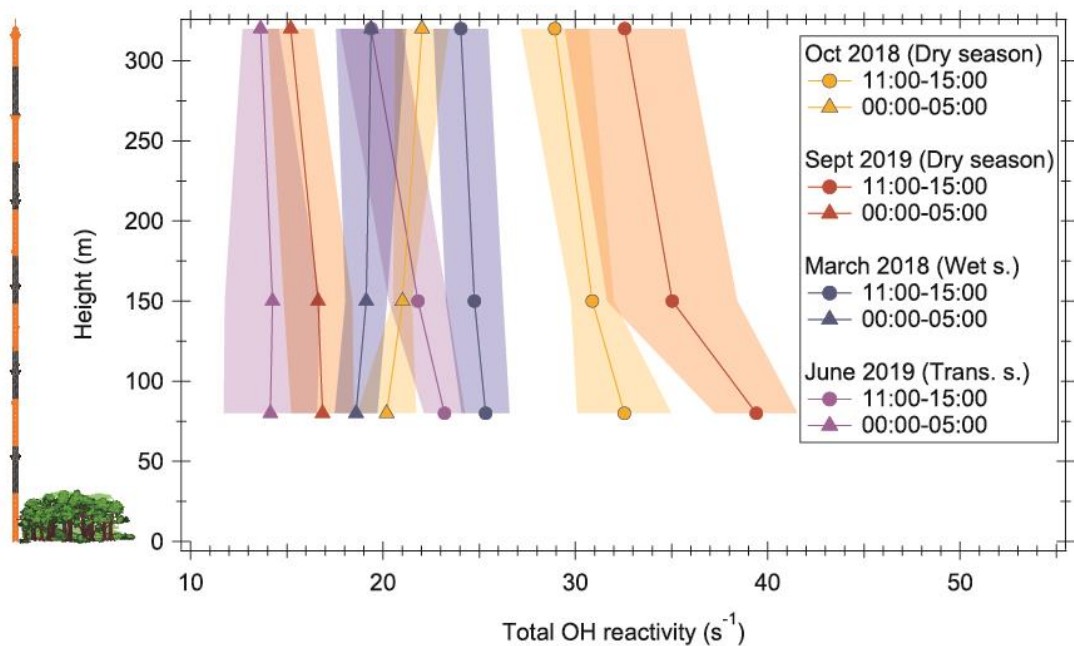

**Figure 1: Profiles of total OH reactivity at the ATTO tower for two dry seasons, one wet and one transition season. Values are averages of several days of measurement (5–13 days, see Sect. 2.2). The shaded areas represent the standard deviation of the data. The left part of the figure illustrates the tower and the canopy height at the same scale as the OH reactivity profiles. Wet s. = wet season, Trans. s. = transition season. All times are local time.**






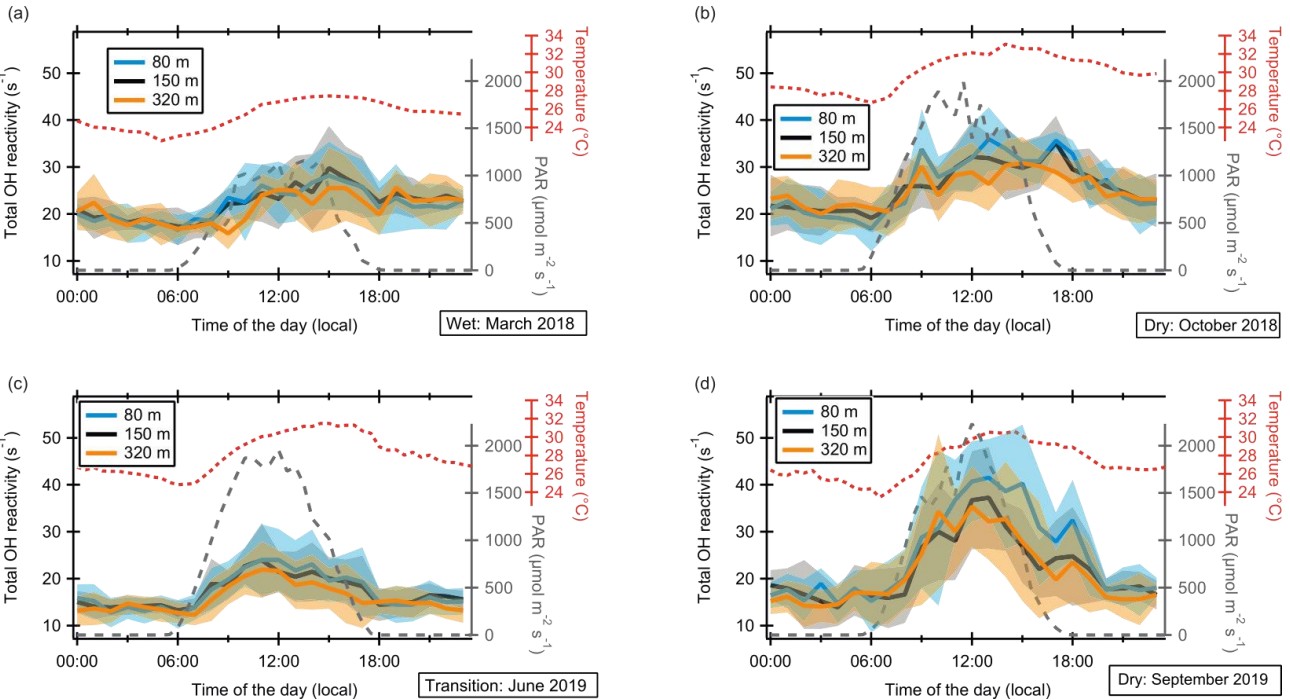

**Figure 2: Diel cycles of total OH reactivity at 80, 150, and 320 m height a. g. l. at the ATTO tower for four measurement periods:**
**(a) March 2018 (wet season), (b) June 2019 (wet–dry transition season), (c) October 2018 (dry season), (d) September 2019 (dry**
**season). Values are hourly diel averages over each measurement period. The shaded areas represent the standard deviation of the**
**data. PAR (photosynthetically active radiation) and temperature were measured at 81 m height a. g. l at the 80 m walk-up tower.**

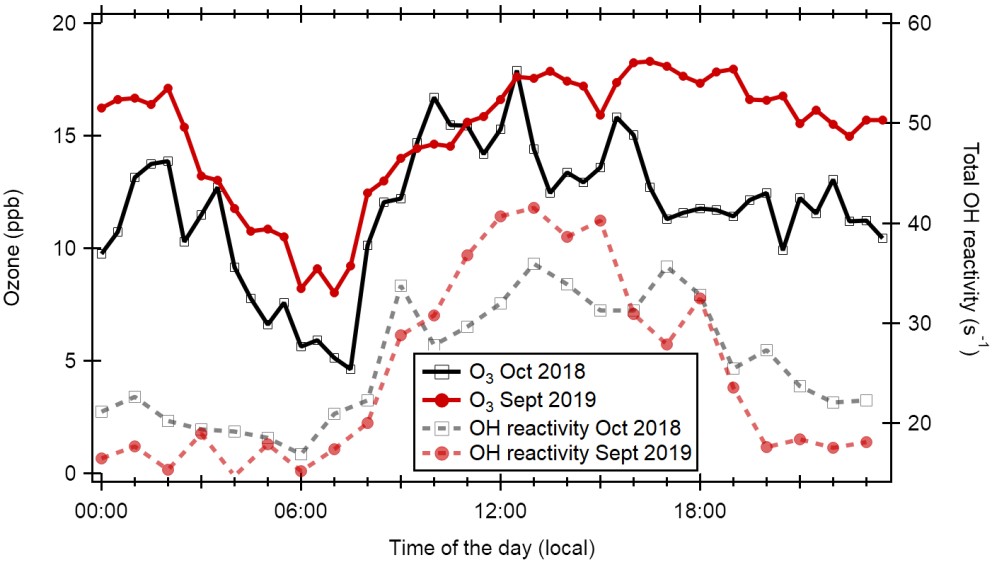

**Figure 3: Diel cycles of ozone mixing ratios (left axis) and total OH reactivity (right axis) at ~ 80 m height a. g. l. for the two dry**
**seasons, October 2018 and September 2019. Values shown are the hourly (reactivity) or half-hourly (ozone) diel averages over the**
**measurement periods.**


**Figure 4: Diel cycles of total OH reactivity, speciated OH reactivity by class of compounds, photosynthetically active radiation (PAR),**
**temperature and unattributed OH reactivity at 80 m above ground level for four measurement periods: (a) March 2018, (b) October**
**2018, (c) June 2019, (d) September 2019. The individual compounds included in each class are listed in Table S1. Values are averages**
**of several days of measurement (5–13 days, see Sect. 2.2). Error bars (for measured OH reactivity) and shaded areas (for**
**unattributed reactivity) represent the standard deviation of the data. The unattributed fraction does not depend on height (see Fig.**
**S2), therefore data from 80 m is shown as a representative example. Note that the axis range of (a) and (b) differs from that of (c)**
**and (d) for better visibility of the reactivity contributions.**





**Figure 5: (a)** Average total OH reactivity and speciation at the ATTO tower by season and height a. g. l., daytime. **(b)** Same as (a), nighttime. **(c)** VOC carbon concentration by compound class by season and height, daytime. **(d)** Same as (c), nighttime. **(e)** Summed VOC mixing ratio (MR) by compound class, season and height, daytime. **(f)** Same as (e), nighttime. The individual compounds included in each class are listed in Table S1. Daytime was defined as PAR > 0 and nighttime as PAR = 0. S: September 2019, O: October 2018, J: June 2019, M: March 2018.





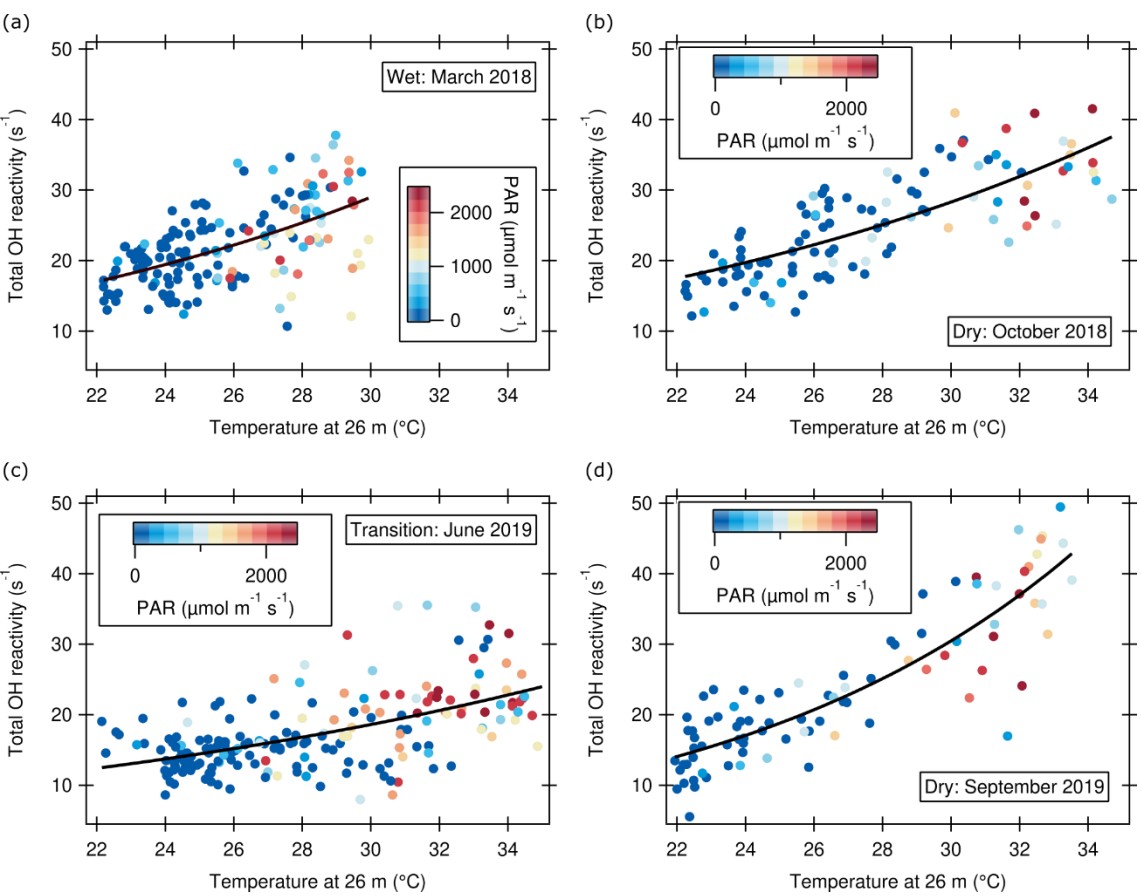

**Figure 6: Hourly averages of total OH reactivity at 80 m a. g. l. at the ATTO tower as a function of temperature at canopy height.**
**(a) Wet season (March 2018), fit function: R = 20.8\*exp(0.066\*[T-25]) (b) Dry season (October 2018), fit function: R = 20.9exp(0.060\*[T-25]) (c) Transition season (June 2019), fit equation: R = 14.4exp(0.051\*[T-25]) (d) Dry season (September 2019), fit equation: R = 18.8exp(0.097\*[T-25]). PAR = photosynthetically active radiation.**



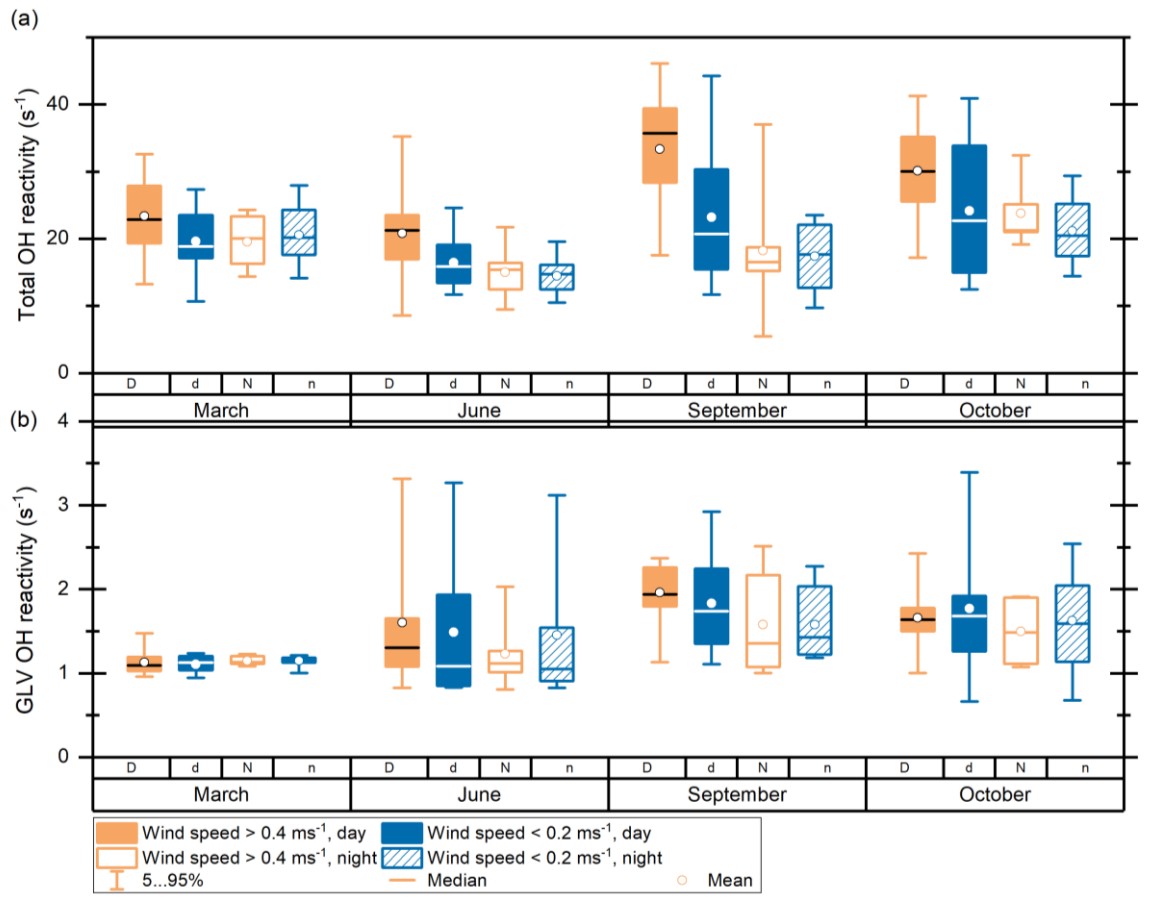

**Figure 7: (a) Total OH reactivity at 80 m a. g. l. at the ATTO tower for two dry (September 2019, October 2018), one wet (March 2018) and one wet-dry transition season (June 2019), categorized by a wind index. Categories: D = wind speed at 26 m > 0.4 m s$^{-1}$, daytime; d = wind speed < 0.2 m s$^{-1}$, daytime; N = wind speed > 0.4 m s$^{-1}$, nighttime, n = wind speed < 0.2 m s$^{-1}$, nighttime. The boxes include the 25th–75th percentiles and the whiskers the 5th–95th percentile. (b) Same as (a), but for calculated OH reactivity of total green leaf volatiles.**

1070



1075

**Figure 8: Total OH reactivity (measured at 80 m a. g. l., hourly averages), wind speed, temperature and precipitation (measured at 320 m, 30 min averages) time series in June 2019 at the ATTO tower. PAR = photosynthetically active radiation, measured at the nearby walk-up tower. (a) Complete time series, (b), (c) and (d) zoomed example rain events, (e) zoomed rain event as in (d), but with OH reactivity calculated from isoprene, monoterpenes, and OVOCs.**

1080


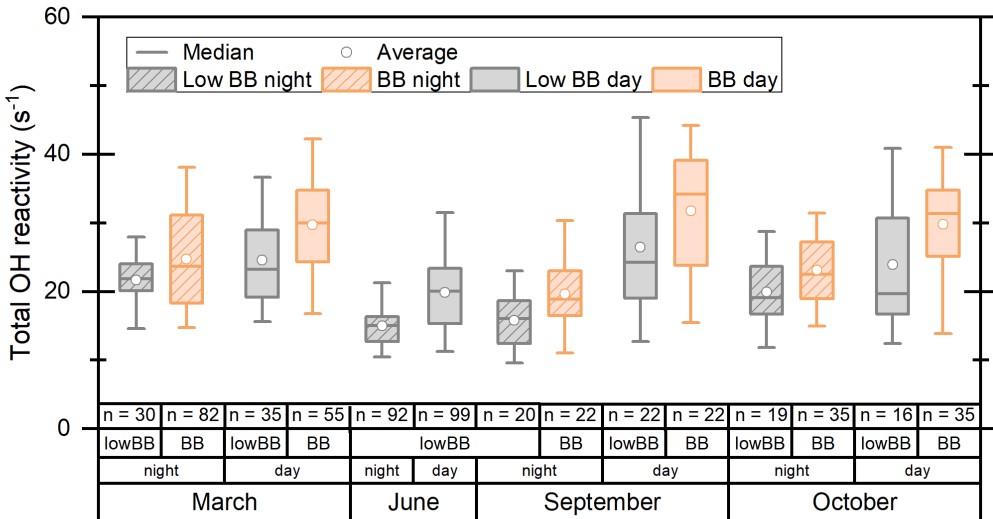

**Figure 9: Total OH reactivity at 80 m a. g. l. at the ATTO tower for two dry seasons (October 2018, September 2019), one wet (March 2018), and one transition season (June 2019), categorized by a biomass burning index. The boxes include the 25th–75th percentiles and the whiskers the 5th–95th percentile. The number of data points in hourly time resolution attributed to each category is shown as "n".**