# Peer review of "Total OH reactivity over the Amazon rainforest: variability with temperature, wind, rain, altitude, time of day, season, and an overall budget closure"

_Atmospheric Chemistry and Physics, 2020_

## Referee Comment (RC1) · Anonymous Referee #1 · 7 Dec 2020

Pfannerstill et al. presented OH reactivity observations at the ATTO tower. The observation was conducted mostly above the canopy in three different levels from 80 m to 300 m AGL. Unlike a previous study presenting significant missing OH reactivity inside of the canopy at the ATTO tower observatory, this study illustrates a better agreement between observed and calculated OH reactivity. Overall, this study concludes that most of reactive compounds in the forest canopy are oxidized before it reaches in the observed altitudes therefore better characterizations in OVOCs are necessary to close OH reactivity budget. The comprehensive dataset is well presented and the data

analysis is easy to follow. Moreover, the conclusion certainly has merits to better understand atmospheric chemistry in remote rain forests, which the community lacks of empirical studies. I would recommend publication of this manuscript after addressing and clarifying following points.

1) Although the contribution of isoprene towards calculated OH reactivity is less in this observation, conducted out of canopy, still isoprene substantially contributes calculated OH reactivity. It seems to me that additional discussion about photochemical aging time scale would be beneficial to make the reasoning of the large contributions of OVOCs towards calculated OH reactivity more convincing by analyzing isoprene to MVK+MACR ratios or some other indicators. I was a bit confused by taking a look at Figure 5 a) that the relative contribution of isoprene does not seem to change too much as observational altitudes get higher. Moreover, if there is any unquantified reactive VOC causing missing OH reactivity inside of the canopy as previously observed, the quantitative analysis may provide clues on the potential contributions of those compounds in the observed altitudes.

2) A more detailed description on sampling, particularly potential sampling loss would be beneficial. I agree that the comparison analysis between observed and calculated OH reactivity was performed for the samples collected from the same inlets so comparison itself is apple to apple comparisons. However, it is certainly possible highly oxidized VOCs or large VOCs such as sesquiterpenes that happen to be soluble and wall reactive may substantially contribute towards calculated OH reactivity. At least, rough estimates based upon empirical proof are highly desirable based upon the wall loss test instead of a short description as presented in the manuscript.

3) A detailed presentation on OVOC speciation would be informative. I would recommend to add more information on relative contributions of each OVOCs and their origins (parent compounds) towards calculated OH reactivity.

4) Figure 6: it is difficult to read out which factor either temperature or PAR (likely both)

would cause trend. I would recommend to chop out certain ranges of PAR to see the temperature dependence.

---

## Referee Comment (RC2) · Anonymous Referee #2 · 14 Jan 2021

General comments: The manuscript Total OH reactivity over the Amazon rainforest: variability with temperature, wind, rain, altitude, time of day, season, and an overall budget closure by Pfannerstill et al (acp-2020-752) addressed a very interesting topic on closure evaluation of OH reactivity. The measurement data obtained at Amazon rainforest were valuable, the methodology for VOC speciation measurement by using PTR_TOF and data processing for investigation variation of OH reactivity were solid and reasonable. The reviewer considers that the detailed study on OH reactivity of this kind are very much needed, and would suggest acceptance for publication after

the following suggestion are considered in revising the current version. Specific comments: 1 I wonder why black carbon measurement was mentioned in this MS which was not used for analysis, and would suggest that the authors provide more discussion on VOCs speciation for OH reactivity closure purpose, e.g. key species that were not measured from previous studies; 2 I believe that the contribution of VOCs groups to total OH reactivity could be very different during normal condition (day and night), precipitation, or biomass burning events. The inter-comparison for major VOCs species attributing to OH reactivity would be important and useful; 3 The major concern was the main parameters influencing OH reactivity and the approach to quantify the parameters. The MS did not provide explanation why OH reactivity varied with precipitation process, and I wonder why authors use only temperature to parameterize OH reactivity from biogenic emissions, fig 6 showed clearly the regressions were not linear for temperature, and MEGEN model quantified already the role of temperature and PAR in VOCs emissions.

---

## Author Comment (AC1) · 4 Feb 2021

We thank the referee for the time invested in reviewing this manuscript, and for their comments which helped improve the new version.

**Reviewer comment 1:**
Although the contribution of isoprene towards calculated OH reactivity is less in this observation, conducted out of canopy, still isoprene substantially contributes calculated
OH reactivity. It seems to me that additional discussion about photochemical
aging time scale would be beneficial to make the reasoning of the large contributions
of OVOCs towards calculated OH reactivity more convincing by analyzing isoprene to
MVK+MACR ratios or some other indicators. I was a bit confused by taking a look
at Figure 5 a) that the relative contribution of isoprene does not seem to change too
much as observational altitudes get higher. Moreover, if there is any unquantified reactive
VOC causing missing OH reactivity inside of the canopy as previously observed,
the quantitative analysis may provide clues on the potential contributions of those compounds
in the observed altitudes.

Response: Thanks for the suggestion to consider isoprene oxidation products vs. isoprene to show more clearly how the VOCs become more oxidized towards the top of the tower. We have now added a figure and a paragraph about isoprene and monoterpene oxidation product/precursor ratios which also shows more clearly that the isoprene contribution is decreasing towards the top. Oxidation timescales can, however, not be derived from this, because this would require knowledge of mixing and transport time scales, which are unfortunately unavailable.

We thought about extrapolating the gradients towards canopy height so that we could potentially draw conclusions on the fractions of OVOCs and their precursors in previous studies, where the measurement height was lower. However, we dismissed this option as too speculative, because we do not know whether the gradients are linear and whether the mixing and turbulence are different directly above the canopy. Indeed, it is expected that the roughness layer extends to 3-4x the canopy height, meaning that the 80 m sampling height would be inside the roughness layer, while 150 m and 320 m are not. Moreover, it has to be noted that the 80 m measurement height in this study overlaps with the highest height included in Nölscher et al. (2016).

Nonetheless, we added a new figure (Fig. 6) and the following paragraph to the manuscript in Sect. 3.2.3:

"Fig. 6 (e) illustrates the increasing photochemical age of air towards the top of the tower by showing ratios of isoprene and monoterpene oxidation products vs. precursors. The monoterpene oxidation products/monoterpenes (MTO/MT) ratios are generally lower than the isoprene oxidation products/isoprene ratios (IsopO/Isop) because the oxidation of isoprene with OH is faster than that of the average monoterpene. For the ATTO monoterpene mixture as identified by TD-GC-ToF-MS (Table S1), the ratio of k(OH + isoprene)/k(OH + monoterpenes) was around 1.38. In agreement with this difference in OH oxidation velocity, the ratio (IsopO/Isop)/(MTO/MT) was between 1.01 (80 m, rainy season) and 2.34 (80 m, dry season).

Figure 6 also shows seasonal variability of the oxidation state and oxidation product/precursor ratios. Fractions of highly oxygenated VOCs (Fig. 6 (a-d)) and oxidation product/precursor ratios (Fig. 6 e) were lowest in the rainy season (March 2018) and higher in the transition (June 2019) and dry seasons (October 2018, September 2019). This seasonality in ratio corresponds to the seasonal differences in solar irradiation with higher OH production rate in the dry season and thus increased photochemistry. The low MTO/MT ratio in September 2019 was due to similar concentrations of oxidation product despite higher monoterpene concentrations than in June 2019."

[Figure]

Figure 6. (a) Vertical profiles of average daytime OH reactivity contribution by general chemical formula ($C_xH_y$, $C_xH_yO$, $C_xH_yO_2$, $C_xH_yO_3$) for both dry seasons. (b) Enlarged lower range of (a) in log scale. (c) Same as (a), but for wet and transition seasons. (d) Enlarged lower range of (c) in log scale. (e) Ratios of oxidation products over their precursors by height above ground level and season. MTO/MT: m/z 139.11 (monoterpene oxidation products) vs. monoterpenes. IsopO/Isop: m/z 71.05 (isoprene oxidation products) vs. isoprene. For October 2018, no monoterpene oxidation product data were available due to issues with peak identification at this ion mass. Data are given for noontime to early afternoon (11:00-15:00).

**Reviewer comment 2:** A more detailed description on sampling, particularly potential sampling loss would be beneficial. I agree that the comparison analysis between observed and calculated OH reactivity was performed for the samples collected from the same inlets so comparison

itself is apple to apple comparisons. However, it is certainly possible highly oxidized VOCs or large VOCs such as sesquiterpenes that happen to be soluble and wall reactive may substantially contribute towards calculated OH reactivity. At least, rough estimates based upon empirical proof are highly desirable based upon the wall loss test instead of a short description as presented in the manuscript.

Response:

The loss of OH reactivity and individual VOCs during sampling can be due to a) chemical loss as a result of reactions with ozone (which is not scrubbed before air enters the inlet line because the same line is used for ozone measurements), and b) reversible loss due to interaction with the Teflon surfaces. Regarding a), we considered the VOCs measured that react fastest with ozone, i.e. sesquiterpenes. The sesquiterpenes observed at 80 m above ground and higher are relatively unreactive, as the most reactive ones never even reach this altitude, but are lost in the upper canopy. The speciation of sesquiterpenes by TD-GC-ToF-MS revealed a-copaene, cyperene, longifolene, and cyclosativene (see Table S1), with a weighted average reaction rate coefficient of $1.23E{-}16$ cm³ molecules$^{-1}$ s$^{-1}$ towards ozone in their most reactive composition (dry season, lowest height). Using this reaction rate coefficient and the highest diel ozone mixing ratio in the dry seasons of ca. 17 ppb, the lifetime of the here observed sesquiterpene mixture towards ozone is 5.5 h, and therewith much longer than the 80 seconds the VOCs spend in the inlet tube. We therefore conclude that loss of VOCs due to ozone reactions in the inlet is negligible.

Regarding b), the reversible loss, literature confirms that inlet tubing acts similarly to a chromatographic column, i.e. that VOCs can adhere to the surfaces temporarily, but are eventually driven through, so that peaks are broadened while actual overall concentration losses are low. This means that the total OH reactivity observed in this study may be underestimated in its peak values and smeared towards later hours of the day, but that the overall daytime sum should be correct. The inlet tubing is constantly flushed even when it is not being sampled, so that surfaces are assumed to be saturated.

We added a new section to the methods to include this information:

**"2.4 Inlet effects**

Due to the length of the inlet tubing of more than 320 m from the top of the ATTO tower to the instrumentation, potential sampling losses have to be considered. Losses can be irreversible due to oxidation with ozone, or reversible due to interactions with the tubing. The most ozone-reactive VOCs observed in this study are sesquiterpenes, and the sesquiterpene mixture at 80 m a.g.l. and higher was relatively unreactive, as can be seen in the TD-GC-ToF-MS-derived speciation in Table S1. The most reactive sesquiterpene mixture observed (lowest height, dry season) had an average reaction rate coefficient of $1.23E{-}16$ cm³ molecules$^{-1}$ s$^{-1}$ towards ozone and thus a lifetime of ca. 5.5 h for peak diel ozone concentrations of 17 ppb (dry season). The inlet residence time of 80 s is therefore not considered to be long enough for significant losses in sesquiterpenes. Potential chemical losses due to ozone being present in the inlet tube are thus considered negligible.

VOCs, especially polar compounds, can partition reversibly from the gas phase to Teflon tubing walls (Deming et al., 2019; Pagonis et al., 2017; Liu et al., 2019). This causes a delay in the time profile of the VOCs, similar to the effect a chromatographic column (Pagonis et al., 2017). In consequence, concentration peaks are smeared and broadened, which causes an underestimation of observed concentrations (Deming et al., 2019). The ATTO inlet tubing was constantly flushed even when not being sampled, so that surfaces were assumed to be saturated or near to equilibrium, so that inlet

interactions are minimized. We determined inlet losses in peak concentrations from the 320 m inlet to the instrument by introducing a calibration gas mixture from the top of the tower. The losses in peak concentration ranged between 11 % and 30 % for all substances in the calibration mixture. We did not correct final data for inlet losses in order to keep VOC and total OH reactivity data, which were both measured from the same inlet, comparable (inlet loss correction for total OH reactivity would change according to its composition, which is not entirely known), and because loss fractions for substances not included in the calibration mixture are not known. This means that the total OH reactivity and VOC concentrations observed in this study may be underestimated in their peak values and smeared towards later times of the day, but that the overall daytime sum should be unaffected.

"

**Reviewer comment 3:** A detailed presentation on OVOC speciation would be informative. I would recommend to add more information on relative contributions of each OVOCs and their origins (parent compounds) towards calculated OH reactivity.

Response: We agree it is a good idea to elaborate on OVOC speciation due to the relevance of these VOCs for the results. A detailed speciation of OVOCs as well as the other compounds can be found in Table S1. As can be seen there, for most OVOCs, the identification is relatively uncertain because several possible structural formulas could be attributed to the corresponding chemical formula. This is also why the identification of potential parent molecules is not feasible. However, we added two figures to illustrate the distribution of the OH reactivity of OVOCs by molecular mass (new Fig. S4) and show vertical profiles by number of oxygens in the molecule and by ratio of oxidation products vs. precursors (new Fig. 6).

We added the following to the manuscript in Sect. 3.2.3:

"In all seasons during daytime, the OH reactivity contribution of directly emitted BVOCs (isoprene, terpenoids) decreased with height (e.g. for isoprene from 51 % at 80 m to 39 % at 320 m in October 2018), while the fraction of oxidation products, i.e. OVOCs, increased (e.g. from 31 % to 39 % in October 2018). Also, the number of oxygen atoms in OVOCs increased with height, while the OH reactivity fraction of non-oxygenated $C_xH_y$ VOCs decreased, as shown in Fig. 6 (a-d). This is due to the increasing average photochemical age of the air mass with height, which increases the abundance of OVOCs relative to primary BVOCs such as isoprene (Karl et al., 2009). Fig. 6 (e) illustrates the increasing photochemical age of air towards the top of the tower by showing ratios of isoprene and monoterpene oxidation products vs. precursors. The monoterpene oxidation products/monoterpenes (MTO/MT) ratios are generally lower than the isoprene oxidation products/isoprene ratios (IsopO/Isop) because the oxidation of isoprene with OH is faster than that of the average monoterpene. For the ATTO monoterpene mixture as identified by TD-GC-ToF-MS (Table S1), the ratio of k(OH + isoprene)/k(OH + monoterpenes) was around 1.38. In agreement with this difference in OH oxidation velocity, the ratio (IsopO/Isop)/(MTO/MT) was between 1.01 (80 m, rainy season) and 2.34 (80 m, dry season).

Figure 6 also shows seasonal variability of the oxidation state and oxidation product/precursor ratios. Fractions of highly oxygenated VOCs (Fig. 6 (a-d)) and oxidation product/precursor ratios (Fig. 6 e) were lowest in the rainy season (March 2018) and higher in the transition (June 2019) and dry seasons (October 2018, September 2019). This seasonality in ratio corresponds to the seasonal differences in solar irradiation with higher OH production rate in the dry season and thus increased

photochemistry. The low MTO/MT ratio in September 2019 was due to similar concentrations of oxidation product despite higher monoterpene concentrations than in June 2019.

„

[Figure]

*Figure S4. Daytime OH reactivity for each season by protonated mass-to-charge ratio (m/z) of PTR-ToF-MS ions and colored by compound class (see Table S1). Isoprenoids include isoprene, monoterpenes and sesquiterpenes as specified in Table S1 (including primarily emitted oxygenated monoterpenes). Ions colored as isoprenoids or GLVs include fragments as denoted in Table S1. Note that the y axis is in logarithmic scale.*

**Reviewer comment 4**:

Figure 6: it is difficult to read out which factor either temperature or PAR (likely both) would cause trend. I would recommend to chop out certain ranges of PAR to see the temperature dependence.

Response: Thanks for this comment. It is true that it is difficult to disentangle PAR and temperature impacts on OH reactivity from one another. The best graphical solution to this problem we found was to also show OH reactivity vs. PAR plots, now in Fig. S5. These illustrate that the correlation of OH reactivity with PAR is weaker than with temperature. We have two reasons to use temperature rather than PAR to parameterize OH reactivity: 1) There is a nighttime temperature dependence of OH reactivity (i.e. see datapoints with PAR = 0) which we could not explain by using a PAR-dependent parameterization. We added plots of OH reactivity vs PAR to the supplement for illustration of the large OH reactivity range at PAR = 0. The correlation of OH reactivity with temperature is, as the comparison with Fig. 7 shows, better than its correlation with PAR. 2) Temperature is, during daytime, strongly influenced by PAR and therefore should be a proxy for it.

Thus, we assume that PAR influences on OH reactant levels at the ATTO tower will be sufficiently captured by using the temperature-dependent parameterization given.

We rephrased the beginning of Sect 3.3.1 as following: "As illustrated in Fig. S5 and by the color scaling in Fig. 7, higher temperature and OH reactivity often co-occurred with higher PAR because temperature is driven by PAR in the daytime. PAR is a driver of reactive emissions in the rainforest (Kuhn et al., 2004a; Jardine et al., 2015). However, there is a PAR-independent temperature dependence visible at PAR = 0, i.e. during the night, and the correlation of OH reactivity with PAR was weaker than with temperature (Fig. 7, Fig. S5). This is why we chose to parameterize OH reactivity based on temperature rather than PAR. Air temperature can serve as a proxy for the combined effects of direct light- and temperature-dependent emission as well as transport, which all influence observed total OH reactivity. Thus, in this simplistic approach, we assume that any PAR- and transport-related influences on OH reactant levels at the ATTO tower will be captured indirectly by using a temperature-dependent parameterization. "

[Figure]

*Figure S5.  Hourly averages of total OH reactivity at 80 m a. g. l. at the ATTO tower as a function of photosynthetically active radiation (PAR). Temperature color scale shown in (b) for all panels. (a) Wet season (March 2018), fit function: R = 21.7– 0.003\*[PAR], r² = 0.08 (b) Transition season (June 2019), fit function: R = 15.4– 0.004\*[PAR], r² = 0.27 (c) Dry seasons (October 2018 and September 2019), fit equation: R = 21.4– 0.006\*[PAR], r² = 0.27.*

---

## Author Comment (AC2) · 4 Feb 2021

We thank the referee for their constructive comments which have helped to improve the manuscript.

**Reviewer comment 1:** I wonder why black carbon measurement was mentioned in this MS which was not used for analysis, and would suggest that the authors provide more discussion on VOCs speciation for OH reactivity closure purpose, e.g. key species that were not measured from previous studies;

Response: The reason why black carbon measurements are mentioned is because black carbon was used to identify the periods influenced by biomass burning as described in Sect. 2.7 and used in Sect. 3.3.4.

The key species not included in previous studies are discussed in Sect. 3.2.1. However, our analysis shows that there are not only few key species that mainly explain the previously missing fraction. Instead, it is rather the sum of many small contributions, in particular from OVOC. For a comprehensive overview of VOCs which were included in the previous study by Nölscher et al., we now added markers to these VOCs in Table S1. The following was added in Sect 3.2.1:

"As a comparison to a previous OH reactivity study at the ATTO site, the compounds included in Nölscher et al. (2016) were marked with a "*" in Table S1."

And

"Our analysis shows that the previously unattributed OH reactivity cannot be explained by few key species, but rather that a multitude of relatively small contributions, in particular from OVOC, explains the sum of OH reactivity. As illustrated in Fig. S4 and Table S1, no VOC exceeds an average OH reactivity of 1 s$^{-1}$ except for isoprene, and its oxidation product measured on m/z = 71 (MVK/MACR/ISOPOOH)."

**Reviewer comment** 2: I believe that the contribution of VOCs groups to total OH reactivity could be very different during normal condition (day and night), precipitation, or biomass burning events. The inter-comparison for major VOCs species attributing to OH reactivity would be important and useful;

Response: Thanks for this suggestion. Day and night differences in speciation were already analyzed in Sect. 3.2.3. However, in light of this comment we have now added an analysis of the biomass burning impact on OH reactivity speciation to Sect. 3.3.4. The following was added to Sect. 3.3.4.: "The OH reactivity speciation differed between biomass burning and low or no biomass burning periods. Under biomass burning influence, the share of OVOCs in total OH reactivity increased on average by 8.1 %, while the isoprenoids fraction decreased by 3.5 %. Biomass burning is a well-known source of both aromatics and nitrogen-containing VOCs [1], which, however, both have low reaction frequencies with OH. Therefore, the OH reactivity fraction of aromatics and nitrogen-containing VOCs increased by only on average 0.05 % and 0.03 %, respectively. "

We were hesitant with showing comparisons of the speciation associated with rain events to "normal" periods because the rain-associated behavior of VOCs is relatively complicated with sharp increases at the beginning of a rain event followed by strong decreases, as shown in Fig. 9 (Fig. 8 in

the previous version of the manuscript). The behavior of OVOCs is discussed in Sect. 3.3.3., with a significant decrease in OVOCs following the rain. However, we now also include the influence of rain events on monoterpenes and isoprene. The following was added to Sect. 3.3.3.:

"When averages of daytime periods without rain are compared to periods associated with daytime rain events, OH reactivity attributed to isoprene and monoterpenes decreased from 36 ± 19 % and 6 ± 4 % of the total, respectively, to 32 ± 18 % and 3 ± 1 %, respectively (Table S2). This can be attributed to the lower irradiation and temperature during rainy periods, which leads to lower primary BVOC emissions from plants. "

The following table was added to the supplement as Table S2:

| OH reactivity contributed by | Rain (daytime, n= 18, ± standard deviation) | Dry (daytime, n=81, ± standard deviation) |
|---|---|---|
| Isoprene | 32 ± 18 % | 36 ± 19 % |
| Monoterpenes | 3 ± 1 % | 6 ± 4 % |
| OVOC | 16 ± 6 % | 22 ± 10 % |
| GLV | 6 ± 2 % | 9 ± 4 % |
| SQT | 1 ± 0.3 % | 1 ± 0.9% |
| Inorganics | 3 ± 0.1 % | 3 ± 0.1 % |
| Others | 0 % | 0 % |
| Unattributed fraction | 39 ± 19 % | 23 ± 25 % |

**Reviewer comment** 3: The major concern was the main parameters influencing OH reactivity and the approach to quantify the parameters. The MS did not provide explanation why OH reactivity varied with precipitation process, and I wonder why authors use only temperature to parameterize OH reactivity from biogenic emissions, fig 6 showed clearly the regressions were not linear for temperature, and MEGEN model quantified already the role of temperature and PAR in VOCs emissions.

Response:

Thanks for this comment. It is true that it is difficult to disentangle PAR and temperature impacts on OH reactivity from one another. The best graphical solution to this problem we found was to also show OH reactivity vs. PAR plots, now in Fig. S5. These illustrate that the correlation of OH reactivity with PAR is weaker than with temperature, and that there is a large range of OH reactivity at PAR=0. We have three reasons to use temperature rather than PAR to parameterize OH reactivity: 1) There is a nighttime temperature dependence of OH reactivity (i.e. see datapoints with PAR = 0) which we could not explain by using a PAR-dependent parameterization. We added plots of OH reactivity vs PAR to the supplement for illustration of the large OH reactivity range at PAR = 0. The correlation of OH reactivity with temperature is, as the comparison with Fig. 7 shows, better than its correlation with PAR. 2) Temperature is, during daytime, strongly influenced by PAR and therefore should be a proxy for it. Thus, we assume that PAR influences on OH reactant levels at the ATTO tower will be sufficiently captured by using a temperature-dependent parameterization. 3) The parameterization

here is not intended to describe emissions, because OH reactivity is a result of VOC concentration levels. It is intended to allow comparison of model generated data with this measurement dataset. Just like the VOC concentrations and not the VOC emission fluxes, OH reactivity follows the diel pattern of temperature, not PAR, in the rainforest, as it is broadened towards the evening [2,3].

The parameterization suggested in our manuscript is by no means intended to replace emission models such as MEGAN. We intended this parameterization as an offer to modelers to compare their results with OH reactivity.

Regarding the rain influence, the wet and transition data show that despite the relatively regular occurrence of rain events, the correlation between OH reactivity and temperature remained strong. This may be because the effect of rain on OH reactivity was very variable as discussed above. The potential reasons for the observed influence of precipitation on OH reactivity was discussed in Sect. 3.3.3. The main effects were suggested to be changes in upwards transport of VOCs due to the convective nature of these events, and a suppression of plant emissions with decreased irradiation and temperature.

We rephrased L. 433ff as following: "As illustrated in Fig. S5 and by the color scaling in Fig. 7, higher temperature and OH reactivity often co-occurred with higher PAR because temperature is driven by PAR at daytime. PAR is a driver of reactive emissions in the rainforest (Kuhn et al., 2004a; Jardine et al., 2015). However, there is a PAR-independent temperature dependence visible at PAR = 0, i.e. during the night, and the correlation of OH reactivity with PAR was weaker than with temperature (Fig. 7, Fig. S5). This is why we chose to parameterize OH reactivity based on temperature rather than PAR. Air temperature can serve as a proxy for the combined effects of direct light- and temperature-dependent emission as well as transport, which all influence observed total OH reactivity. Thus, in this simplistic approach, we assume that any PAR- and transport-related influences on OH reactant levels at the ATTO tower will be captured indirectly by using a temperature-dependent parameterization. The relationship is not intended to be used as an emission algorithm but to facilitate comparison with model generated results. "

[Figure]

**Figure S1. Hourly averages of total OH reactivity at 80 m a. g. l. at the ATTO tower as a function of photosynthetically active radiation (PAR). Temperature color scale shown in (b) for all panels. (a) Wet season (March 2018), fit function: R = 21.7– 0.003\*[PAR], $r^2$ = 0.08 (b) Transition season (June 2019), fit function: R = 15.4– 0.004\*[PAR], $r^2$ = 0.27 (c) Dry seasons (October 2018 and September 2019), fit equation: R = 21.4– 0.006\*[PAR], $r^2$ = 0.27.**

**References**

[1] A.R. Koss, K. Sekimoto, J.B. Gilman, V. Selimovic, M.M. Coggon, K.J. Zarzana, B. Yuan, B.M. Lerner, S.S. Brown, J.L. Jimenez, J. Krechmer, J.M. Roberts, C. Warneke, R.J. Yokelson, J. de Gouw, Non-methane organic gas emissions from biomass burning: identification, quantification, and emission factors from PTR-ToF during the FIREX 2016 laboratory experiment, Atmospheric Chemistry and Physics 18 (2018) 3299–3319. https://doi.org/10.5194/acp-18-3299-2018.

[2] C. Sarkar, A.B. Guenther, J.-H. Park, R. Seco, E. Alves, S. Batalha, R. Santana, S. Kim, J. Smith, J. Tóta, O. Vega, PTR-TOF-MS eddy covariance measurements of isoprene and monoterpene fluxes from an eastern Amazonian rainforest, Atmos. Chem. Phys. 20 (2020) 7179–7191. https://doi.org/10.5194/acp-20-7179-2020.

[3] U. Kuhn, M.O. Andreae, C. Ammann, A.C. Araújo, E. Brancaleoni, P. Ciccioli, T. Dindorf, M. Frattoni, L.V. Gatti, L. Ganzeveld, B. Kruijt, J. Lelieveld, J. Lloyd, F.X. Meixner, A.D. Nobre, U. Pöschl, C. Spirig, P. Stefani, A. Thielmann, R. Valentini, J. Kesselmeier, Isoprene and monoterpene fluxes from Central Amazonian rainforest inferred from tower-based and airborne measurements, and implications on the atmospheric chemistry and the local carbon budget, Atmos. Chem. Phys. 7 (2007) 2855–2879. https://doi.org/10.5194/acp-7-2855-2007.

---

## Author Response (AR2)

Dear Laurens,

Thank you for reviewing our manuscript so carefully. We revised it according to your suggestions:

1) *You propose this parameterization of OH reactivity using Temperature as the main dependent parameter rather than PAR; this feature is also brought up by the reviewers and have carefully read your response; you indicate that there is also a whole range of significant OH reactivity measurements at night when PAR=0 and which seems to result in a less strong correlation between OH reactivity and PAR than between reactivity and T. If was though wondering if you actually considered, also based on some of the known causal links here between OH reactivity and PAR to only use the daytime data on PAR and OH reactivity and split this from the nocturnal OH reactivity data and how these correlate with T?*

We followed your proposition and tried the parameterization only using daytime data. The dependence of OH reactivity on PAR did not become much more visible by using only daytime data, the correlations are very weak (r2 between 0.01 and 0.14) and the slope of the fit is, especially in the wet season, barely above zero:

[Figure]

**Fig. Hourly averages of total OH reactivity at 80 m a. g. l. at the ATTO tower as a function of photosynthetically active radiation (PAR). Temperature color scale shown in (b) for all panels. (a) Wet season (March 2018), fit function: $R = 22.6 + 0.0011*[PAR]$, $r^2 = 0.01$ (b) Transition season (June 2019), fit function: $R = 16.7 + 0.0027*[PAR]$, $r^2 = 0.12$ (c) Dry seasons (October 2018 and September 2019), fit equation: $R = 24.2 + 0.005*[PAR]$, $r^2 = 0.14$.**

This behavior reflects the trends shown in the diel cycle plots, where OH reactivity lags approximately one hour behind PAR. Then, we also tested how much the fit changes for OH reactivity in dependence of temperature when only using daytime data. See the following two tables:

 **Table 2. Parameterization of total OH reactivity in dependence of temperature: Coefficients for the equation $R = R_s \exp(\beta[T-T_s])$, with $R$ = total OH reactivity, $T$ = temperature in °C, $T_s$ = standard temperature (25 °C), $R_s$ = total OH reactivity at standard temperature, and $\beta$ is an empirical coefficient.**

| Season | $R_s$ | $\beta$ | $r^2$ of fit |
|---|---|---|---|
| Dry | 19.9 ± 0.4 | 0.076 ± 0.004 | 0.68 |
| Wet | 20.8 ± 0.4 | 0.066 ± 0.008 | 0.33 |
| Transition | 14.4 ± 0.4 | 0.051 ± 0.005 | 0.33 |

**NEW Parameterization of total OH reactivity in dependence of temperature, daytime data only: Coefficients for the equation $R = R_s \exp(\beta[T-T_s])$, with $R$ = total OH reactivity, $T$ = temperature in °C, $T_s$ = standard temperature (25 °C), $R_s$ = total OH reactivity at standard temperature, and $\beta$ is an empirical coefficient.**

| Season | $R_s$ | $\beta$ | $r^2$ of fit |
|---|---|---|---|
| Dry | 20.3 ± 0.9 | 0.072 ± 0.007 | 0.60 |
| Wet | 20.1 ± 0.8 | 0.074 ± 0.013 | 0.31 |
| Transition | 14.9 ± 0.9 | 0.050 ± 0.008 | 0.30 |

The fit functions did not change significantly (all changes are within the uncertainty of the previous, day and night, fits). Also, the r2 values decreased, which is not surprising, because the nighttime data showed the same temperature dependent trends as the daytime data. Therefore, we consider the nighttime data relevant for the parameterization and would rather not change this in the manuscript.

*2) Reading over the discussion about the connection between OH reactivity and the role of rain but also that nocturnal measurements indicate a generally a profile on OH reactivity expressing the potentially important role of in-canopy deposition, I checked out about including a reference to the study by Yanez Serrano, ACP, 2018, https://doi.org/10.5194/acp-18-3403-2018 on analysis of seasonal and diurnal cycles in monoterpenes at the ATTO site. This paper also discusses about the potentially important role of in-canopy removal of these monoterpenes also as a function of canopy wetness, etc. Some of those findings seem to directly connected to the study you present in your paper.*

Thank you for pointing out the related findings in the 2018 Yanez-Serrano paper, which we added as a reference accordingly:

In the section on vertical profiles (3.1.1)

"A study conducted at the ATTO site found strong nighttime canopy deposition of monoterpenes which was a function of surface wetness (Yáñez-Serrano et al., 2018). Deposition can thus partly explain the weaker vertical gradients in the wet season, when the overcast conditions led to weaker irradiation and therewith lower OH levels, while deposition continued or was even increased due to precipitation."

And in the section on precipitation effects (3.3.3):

Stable thermal stratification close to the ground can stop soil emissions from reaching higher altitudes in the rainforest (Kruijt et al., 2000; Gerken et al., 2017; Santana et al., 2018; Pfannerstill et al., 2018), and it has been shown that in-canopy chemistry already strongly decreases reactant levels before they reach the canopy top (Jardine et al., 2011; Jardine et al., 2015; Bourtsoukidis et al., 2018; Yáñez-Serrano et al., 2018).

"Additionally, terpenes can be subject to deposition on wet rainforest surfaces (Yáñez-Serrano et al., 2018)."